# Evidential DualU-Net: Single-Pass Uncertainty for Cell Instance Segmentation

**David Anglada-Rotger** [iD]              DAVID.ANGLADA@UPC.EDU
**Ferran Marques**                  FERRAN.MARQUES@UPC.EDU
**Montse Pardàs**                  MONTSE.PARDAS@UPC.EDU
*Image Processing Group (GPI), Universitat Politècnica de Catalunya (UPC), Barcelona, Spain*

**Editors:** Accepted for publication at MIDL 2026

## Abstract

Accurate and trustworthy cell instance segmentation requires models that not only detect and classify nuclei but also communicate how much evidence supports each prediction. DualU-Net is a fast and effective two-head multi-task architecture for this problem, but—like most deterministic models—it provides no principled uncertainty estimates. We introduce *Evidential DualU-Net*, the first evidential framework for multi-task cell instance segmentation.Its segmentation head predicts Dirichlet concentration parameters, enabling single-pass, closed-form aleatoric, epistemic, and vacuity uncertainties at the pixel level, with instance-level quantities obtained via size-invariant pooling of pixel evidence. The centroid decoder is complemented with two lightweight geometric uncertainty cues that quantify localisation reliability without auxiliary models or sampling. Together, these evidential and geometric measures expose complementary failure modes and allow principled filtering of low-confidence nuclei. Across multi-tissue and multi-stain datasets, Evidential DualU-Net matches or surpasses deep ensembles and MC Dropout in error separation at a fraction of the cost, maintains or improves calibration over deterministic baselines, and generalises across datasets without retuning. This work provides an interpretable and computationally practical uncertainty formulation for digital pathology. Code and weights are available at: https://github.com/davidanglada/Evidential-DualU-Net.

**Keywords:** Cell Instance Segmentation, Evidential Deep Learning, Uncertainty Estimation, Multitask Learning, Nuclei Segmentation

## 1. Introduction

Digital pathology has rapidly expanded with the adoption of whole-slide imaging and large annotated datasets, enabling computational models to support routine diagnostic workflows. Cell-level quantification—including nucleus detection, instance segmentation, and phenotype classification—is central to tasks such as Ki-67 assessment, immune profiling, and tumour microenvironment analysis, yet remains time-consuming and variable when performed manually. Fast and reliable automated systems are therefore essential. Within this context, DualU-Net (Anglada-Rotger et al., 2025), developed inside the DigiPatICS project (Temprana-Salvador et al., 2022) from the Institut Català de la Salut (ICS) of Catalunya, was designed as a lightweight, single-pass architecture capable of accurate and efficient cell instance segmentation.

Beyond accuracy, reliability is equally crucial: *a model should know what it does not know.* Uncertainty estimation is particularly important in digital pathology, where ambiguous morphology, heterogeneous staining, artefacts, and domain shifts routinely lead to

failure cases that must be flagged for review. Crucially, uncertainty must be available *at inference time* without the computational burden of ensembles or sampling-based methods. Existing approaches for uncertainty estimation often require multiple forward passes, making them impractical for high-throughput clinical pipelines. Evidential Deep Learning (EDL) offers a promising alternative, providing closed-form aleatoric, epistemic, and vacuity estimates from a single prediction. However, prior work has focused almost exclusively on semantic segmentation; no existing method provides interpretable, instance-level evidential uncertainty or applies EDL to multi-task cell instance segmentation. Furthermore, current uncertainty methods for instance segmentation remain complex and computationally demanding, underscoring the need for simpler and more principled formulations.

Up to the authors knowledge, this work introduces the first evidential approach to multi-task cell instance segmentation, enabling instance-level uncertainty derived from pixel-level evidence. Our key contributions are: i) we extend DualU-Net with a Dirichlet-based evidential segmentation head and a multi-term loss, producing calibrated aleatoric, epistemic, and vacuity estimates at both pixel and instance level, while preserving the previous segmentation and classification performance; ii) we introduce simple, closed-form geometric uncertainty cues for the centroid decoder (peak and mass), enabling reliable detection-oriented uncertainty without auxiliary models or sampling; iii) through extensive evaluation on multi-tissue and multi-stain histopathology datasets, we show that the proposed evidential scheme matches or improves deep ensembles and Monte Carlo Dropout in error separation at a fraction of the computational cost, generalises across datasets without retuning, and yields interpretable uncertainty maps that clearly expose classification and detection issues relevant for digital pathology workflows.

## 2. Related Work

**Cell instance segmentation:** HoVer-Net (Graham et al., 2019) established the dominant multi-decoder paradigm for nuclear instance segmentation by jointly predicting semantic masks, horizontal/vertical offset maps, and cell-type labels. Transformer-based adaptations such as CellViT (Hörst et al., 2024) and HistoNext (Chen et al., 2025a) retain this multi-head structure while incorporating long-range contextual modelling to refine boundaries and improve classification accuracy, highlighting the effectiveness of combining semantic and detection cues for reliable cell delineation. In our previous work, DualU-Net (Anglada-Rotger et al., 2025) streamlines this design to only two decoders: a semantic segmentation head and a centroid regression head. The centroid head predicts a continuous Gaussian density map centred at each nucleus, constructed during training using a fixed standard deviation $\sigma$ that reflects the expected nucleus scale in the dataset (Xie et al., 2018). At inference, instance segmentation is obtained by combining both decoder outputs through a marker-controlled watershed procedure. Local maxima are first extracted from the predicted Gaussian centroid map and used as instance markers. These markers are then propagated over the semantic segmentation mask using the watershed algorithm, yielding a partition of the foreground into individual cell instances.

**Uncertainty estimation and calibration:** Predictive uncertainty in deep learning usually decomposes into *aleatoric* uncertainty, arising from intrinsic ambiguity in the data, and *epistemic* uncertainty, reflecting limited model knowledge or out-of-distribution behavior

(Kendall and Gal, 2017). Estimating both components simultaneously remains difficult in many tasks. Multi-pass methods such as MC Dropout (MCD) (Gal and Ghahramani, 2016) or deep ensembles (DE) (Lakshminarayanan et al., 2017) provide good approximations of epistemic uncertainty, with the latter shown to remain robust under distribution shift (Ovadia et al., 2020), but they are computationally expensive for day-to-day diagnostic workflows and do not yield explicit aleatoric estimates. Probabilistic segmentation frameworks such as Probabilistic U-Net (Kohl et al., 2018) or PhiSeg (Baumgartner et al., 2019) introduce latent sampling or generative priors and can capture ambiguity, yet they require multiple stochastic passes and are not well suited to densely packed nuclei. None of these approaches provide simple, closed-form estimates of both uncertainty types. Uncertainty has also been investigated for error prediction and active learning in biomedical imaging (Tan et al., 2025b; Anglada-Rotger et al., 2024), though most efforts remain in semantic or single-task settings.

Calibration is equally important, as cross-entropy-trained models often produce overconfident predictions. Post-hoc techniques such as temperature scaling (Guo et al., 2017) adjust confidence after training, while train-time strategies (e.g., MMCE (Kumar et al., 2018), focal-loss variants (Mukhoti et al., 2020) or BSCE-GRA (Lin et al., 2025)) aim to regularize confidence throughout optimization. Despite these advances, calibrated and instance-aware uncertainty estimation for multi-task cell segmentation remains under-explored.

**Evidential Deep Learning (EDL):** EDL introduces a probabilistic view of classification in which the network does not output a single categorical distribution, but instead predicts the parameters of a istribution over categorical distributions. In a standard setting, a categorical likelihood for an input $x$ with class probabilities $\mathbf{p} = (p_1, \ldots, p_K)$ is $p(y = k \mid \mathbf{p}) = p_k$, with $\mathbf{p}$ typically produced by a softmax layer. EDL generalizes this by placing a Dirichlet prior over $\mathbf{p}$. Following Sensoy et al. (Sensoy et al., 2018), the network outputs non-negative evidence values $e_k$, which define concentration parameters $\alpha_k = e_k + 1$ of a Dirichlet distribution $D(\mathbf{p} \mid \boldsymbol{\alpha})$. The predictive probabilities are given by the Dirichlet mean (see Section 3). The Dirichlet formulation allows uncertainty to be read directly from the predicted parameters $\boldsymbol{\alpha}$. The total evidence $S = \sum_k \alpha_k$ reflects how strongly the model supports its prediction: when $S$ is small, the Dirichlet distribution is broad, indicating that the model has not accumulated enough evidence to commit to any class. This behaviour is captured by vacuity, which represents uncertainty due purely to a lack of support in the data. In contrast, the spread of the Dirichlet around its mean captures the remaining uncertainty and gives rise to analytic measures of aleatoric and epistemic uncertainty. All these quantities are obtained in closed form, allowing EDL to produce calibrated uncertainty estimates from a single forward pass without sampling or ensembles. Training encourages the model to increase evidence when predictions are correct and suppress it when they are wrong, preventing unwarranted confidence.

EDL has also been explored in semantic segmentation. In (Ancha et al., 2024) evidential models are applied to pixelwise OOD-aware segmentation. EDL has been also used in several biomedical tasks, such as semantic segmentation (Tan et al., 2025a), uncertainty-guided 3D mitochondria segmentation (Shi et al., 2024), interpretable evidential uncertainty supervision (Li et al., 2025), or semi-supervised segmentation via mutual evidential learning (He et al., 2025). These works demonstrate growing interest in evidential segmentation, but

they remain limited to single-task semantic settings: none provide interpretable uncertainty at the instance level, nor do they extend evidential modeling to multi-task formulations. Recent works have also critically examined the theoretical foundations of evidential deep learning, questioning whether Dirichlet-based uncertainty measures should be interpreted as faithful Bayesian epistemic and aleatoric uncertainty estimates (Shen et al., 2024). In line with these findings, we treat the evidential outputs in this work as practically useful uncertainty proxies rather than strictly probabilistic quantities, and focus on their empirical ability to correlate with model errors at pixel and instance level.

## 3. Materials and Methods

**Datasets.** We evaluate on two annotated histopathology datasets. PanNuke (Gamper et al., 2020) provides 7904 H&E patches ($256 \times 256$) across 19 tissues with 189k nuclei labeled into five classes. We also use a proprietary breast Ki-67 IHC dataset (Anglada-Rotger et al., 2024) with 52 tiles ($1024 \times 1024$) from four patients, each containing pixel-level nuclei masks and three-class labels (positive, negative, non-epithelial). Both datasets are extracted at $40\times$ magnification with a spatial resolution of approximately $0.25\,\mu$m/pixel.

**Evidential segmentation head and loss.** DualU-Net (Anglada-Rotger et al., 2025) contains two decoders: a semantic segmentation head and a centroid-regression head. We keep this architecture but replace the segmentation logits with Dirichlet evidence. For each pixel $x$, the segmentation decoder outputs non-negative evidence values $e_k(x) \geq 0$, which define the Dirichlet concentration parameters $\alpha_k(x) = e_k(x) + 1$, $\boldsymbol{\alpha}(x) = (\alpha_1(x), \ldots, \alpha_K(x))$, the predictive class probabilities are given by the Dirichlet mean $\hat{p}_k(x) = \frac{\alpha_k(x)}{S(x)}, S(x) = \sum_{j=1}^{K} \alpha_j(x)$. The predictive categorical distribution at pixel $x$ is defined as $\hat{\mathbf{p}}(x) = (\hat{p}_1(x), \ldots, \hat{p}_K(x))$. Following (Sensoy et al., 2018), the evidential loss combines a data-fitting term encouraging $\hat{\mathbf{p}}(x)$ to match the one-hot label $\mathbf{y}(x)$ with a KL regularizer that discourages unwarranted evidence. To penalize evidence for incorrect classes while leaving the correct class unpenalized, we construct the modified Dirichlet parameter vector $\tilde{\boldsymbol{\alpha}}(x) = \big(\tilde{\alpha}_1(x), \ldots, \tilde{\alpha}_K(x)\big)$, where each component is defined as

$$\tilde{\alpha}_k(x) = \begin{cases} 1, & \text{if } k = y(x), \\ \alpha_k(x), & \text{otherwise.} \end{cases} \tag{1}$$

This way, the per-pixel segmentation loss is

$$\mathcal{L}_{\text{EDL}}^{\text{seg}}(x) = \|\mathbf{y}(x) - \hat{\mathbf{p}}(x)\|_2^2 + \lambda_{KL} \,\text{KL}\Big(D(\mathbf{p} \mid \tilde{\boldsymbol{\alpha}}(x)) \,\|\, D(\mathbf{p} \mid \mathbf{1})\Big), \tag{2}$$

As shown in (Tan et al., 2025b), incorporating a Dice term improves the optimization dynamics of evidential semantic segmentation. For this reason, all our experiments include an additional Dice component. In the original DualU-Net (Anglada-Rotger et al., 2025), the Dice was class-weighted to mitigate strong label imbalance; however, such weighting is uncommon in EDL frameworks. We therefore evaluate two variants: (i) standard (unweighted) Dice and (ii) class-weighted Dice. The centroid decoder and its regression objective remain unchanged from the original DualU-Net. The full training objective is

$$\mathcal{L} = \lambda_{seg}\mathcal{L}_{\text{EDL}}^{\text{seg}} + \lambda_{dice}\,\mathcal{L}_{\text{Dice}} + \lambda_{cent}\,\mathcal{L}_{\text{cent}}, \tag{3}$$

**Segmentation-head evidential uncertainty.** Let $\mathcal{D}$ be the training dataset and $\hat{y}$ the categorical prediction at pixel $x$, modeled as a random variable $\hat{y} \sim \mathrm{Cat}(\mathbf{p})$ where $\mathbf{p}$ is drawn from the Dirichlet distribution $D(\mathbf{p} \mid \boldsymbol{\alpha}(x))$. For a Bayesian classifier with Dirichlet-distributed class probabilities $\mathbf{p} \sim D(\mathbf{p} \mid \boldsymbol{\alpha}(x))$, as in (Kendall and Gal, 2017; Tan et al., 2025a), we use $u_{\mathrm{ale}}(x) = \mathbb{E}_{\mathrm{Dir}}\big[\mathrm{Var}_{\mathrm{Cat}}\big(\hat{y} \mid \mathbf{p}\big)\big]$, $u_{\mathrm{epi}}(x) = \mathrm{Var}_{\mathrm{Dir}}\big(\mathbb{E}_{\mathrm{Cat}}\big[\hat{y} \mid \mathbf{p}\big]\big)$

For the Dirichlet prior, it admits the following closed forms (see Appendix A):

$$u_{\mathrm{ale}}(x) = \sum_{k=1}^{K} \frac{\alpha_k(x)\big(S(x) - \alpha_k(x)\big)}{S(x)\big(S(x) + 1\big)}, \qquad u_{\mathrm{epi}}(x) = \sum_{k=1}^{K} \frac{\alpha_k(x)\big(S(x) - \alpha_k(x)\big)}{S^2(x)\big(S(x) + 1\big)}. \qquad (4)$$

A third quantity naturally arises in evidential models: vacuity. While aleatoric and epistemic uncertainties separate noise from model uncertainty, vacuity measures the absence of evidence accumulated from the data $u_{\mathrm{vac}}(x) = \frac{K}{S(x)}$.

Cell analysis requires uncertainty not only at the pixel level but also at the instance level, since downstream evaluation (detection F1, classification F1) and clinical interpretation are performed per nucleus rather than per pixel. Instance masks $\Omega_i$ are obtained with the same watershed reconstruction as in DualU-Net (see 2). In evidential classification, Dirichlet parameters are commonly interpreted as accumulated evidence arising from independent observations, in which case evidence is additive in the underlying Gamma space. Under this interpretation, each pixel prediction can be viewed as providing a local Dirichlet evidence vector over classes. If pixel-level evidences were conditionally independent samples of the same latent instance-level variable, a principled Bayesian aggregation would correspond to summing Dirichlet parameters across pixels. However, pixel-level predictions within a nucleus are not independent: they are spatially correlated, share receptive fields, and are influenced by common morphological context. Moreover, nucleus size varies substantially, so summing evidences would cause the total concentration $S$ to scale with instance area, artificially suppressing epistemic uncertainty and vacuity for larger nuclei. Therefore, for each instance, we therefore aggregate evidential parameters by averaging:

$$\bar{\alpha}_k^{(i)} = \frac{1}{|\Omega_i|} \sum x \in \Omega_i \alpha_k(x), \qquad \bar{S}^{(i)} = \sum_{k=1}^{K} \bar{\alpha}_k^{(i)}. \qquad (5)$$

This operation should be understood as a pooling of correlated pixel-level evidence rather than as a Bayesian evidence fusion rule. Averaging preserves the relative evidence proportions learned by the network while enforcing size invariance across instances, yielding a stable instance-level evidence profile from which uncertainty quantities can be consistently derived. An ablation study comparing mean, sum, and median pooling for instance-level aggregation is presented in Appendix C.

At the pixel level, all Dirichlet parameters—including the background class—contribute to uncertainty because they shape the full predictive distribution. However, for instance-level uncertainty we are interested only in the reliability of the classification of a segmented nucleus. Therefore, when computing instance-level uncertainty, we exclude the background component from $\bar{\boldsymbol{\alpha}}^{(i)}$ and renormalize over the $K-1$ foreground classes. This ensures that $u_{\mathrm{ale}}(\Omega_i)$, $u_{\mathrm{epi}}(\Omega_i)$, and $u_{\mathrm{vac}}(\Omega_i)$ quantify uncertainty about the nucleus class, not about residual background evidence. Substituting the resulting foreground-only $\bar{\boldsymbol{\alpha}}^{(i)}$ into $u_{epi}$,

$u_{ale}$, and $u_{vac}$ yields instance-level $u_{ale}(\Omega_i)$, $u_{epi}(\Omega_i)$, and $u_{vac}(\Omega_i)$. To make all uncertainty quantities directly comparable and easily interpretable, we normalize $u_{ale}$, $u_{epi}$, and $u_{vac}$ to the range $[0, 1]$. Each expression admits a closed-form theoretical minimum and maximum determined by the Dirichlet parameters $\boldsymbol{\alpha}$ (see Appendix B). For each uncertainty type, we compute its attainable bounds and apply an affine normalization.

**Centroid-head uncertainty.** While Kendal and Gal (Kendall and Gal, 2017) provide a standard probabilistic framework for regression by minimizing the Gaussian Negative Log Likelihood (NLL), we explicitly opt for a geometric approach. The centroid regression head itself follows the original DualU-Net formulation, without any architectural modification. In sparse centroid regression, the NLL objective is not only prone to optimization instability due to class imbalance, but it also strictly models pixel-intensity noise. In contrast, our proposed geometric reliability measures target structural failures.

Let $g : \mathcal{X} \to [0, \infty)$ denote the Gaussian density map predicted by the centroid decoder, where $g(x)$ is the value at pixel $x$. For each reconstructed nucleus instance $\Omega_i \subset \mathcal{X}$, assumed to arise from an isotropic Gaussian with standard deviation $\sigma$, the ideal density integrates to the analytic mass $G_{max} = 2\pi\sigma^2$. Departures of $g$ from this template reflect unreliable centroid localisation. We extract two complementary geometric cues: (i) *Peak uncertainty*, which assesses the sharpness of the predicted Gaussian by the maximum value $p_{max}^{(i)} = \max_{x \in \Omega_i} g(x)$; diffuse or weak responses indicate uncertain detections. We define

$$u_{peak}(\Omega_i) = 1 - p_{max}^{(i)}. \tag{6}$$

(ii) *Mass-ratio uncertainty*, which measures energy preservation. Let $m_{pred}^{(i)} = \sum_{x \in \Omega_i} g(x)$ denote the predicted mass; deviations from $G_{max}$ are quantified symmetrically as

$$u_{mass}(\Omega_i) = \frac{\left| m_{pred}^{(i)} - G_{max} \right|}{G_{max}}. \tag{7}$$

Values near zero correspond to correct centroid strength, whereas large deviations signal missing, diffuse, or overly dominant Gaussian responses. These two cues provide simple and direct measures of centroid reliability for each nucleus. A single scalar uncertainty value is obtained via a linear combination $u_{cent}(\Omega_i) = \lambda_{peak} \, u_{peak}(\Omega_i) + \lambda_{mass} \, u_{mass}(\Omega_i)$,.

**Two uncertainties for two error types.** For each nucleus $\Omega_i$, our method outputs two complementary uncertainty families. Segmentation-head evidential uncertainties $\left( u_{epi}(\Omega_i), u_{ale}(\Omega_i), u_{vac}(\Omega_i) \right)$ reflect ambiguity in the class distribution and are therefore linked to *classification* errors. Centroid-based geometric scores $\left( u_{cent}(\Omega_i), u_{peak}(\Omega_i), u_{mass}(\Omega_i) \right)$ capture the sharpness and stability of the predicted Gaussian response, making them indicative of *detection* errors. Together, they offer complementary, instance-level reliability signals.

## 4. Results

**Experiments and implementation details.** We follow PanNuke three-fold cross validation and Ki-67 leave-one-patient-out cross validation. Following the original DualU-Net training scheme, ll models use a ResNeXt-50 32×4d (Xie et al., 2016) encoder and Gaussian centroid maps are generated using a fixed standard deviation $\sigma = 5$. Starting from

Table 1: Quantitative performance comparison on PanNuke and Ki-67. Per-class F1 scores are reported for dataset-specific semantic classes, together with instance-level Detection F1 (Det.) and segmentation Dice.

| Model | Classification and Detection ↑ | | | | | | Segmentation ↑ |
|---|---|---|---|---|---|---|---|
| | \multicolumn PanNuke | | | | | | |
| | Neo. | Epi. | Inflam. | Conn. | Dead | Det. | Dice |
| Ours | $0.667_{\pm 0.007}$ | $0.675_{\pm 0.002}$ | $0.557_{\pm 0.016}$ | $0.494_{\pm 0.007}$ | $0.001_{\pm 0.002}$ | $0.799_{\pm 0.002}$ | $0.753_{\pm 0.005}$ |
| Ours w | $0.663_{\pm 0.010}$ | $0.649_{\pm 0.030}$ | $0.559_{\pm 0.002}$ | $0.482_{\pm 0.009}$ | $0.144_{\pm 0.031}$ | $\mathbf{0.812}_{\pm 0.003}$ | $0.761_{\pm 0.007}$ |
| Base | $0.666_{\pm 0.014}$ | $0.680_{\pm 0.004}$ | $0.575_{\pm 0.011}$ | $0.521_{\pm 0.006}$ | $0.243_{\pm 0.172}$ | $0.798_{\pm 0.002}$ | $0.755_{\pm 0.007}$ |
| DE | $\mathbf{0.687}_{\pm 0.014}$ | $\mathbf{0.705}_{\pm 0.011}$ | $\mathbf{0.594}_{\pm 0.009}$ | $\mathbf{0.542}_{\pm 0.002}$ | $\mathbf{0.382}_{\pm 0.051}$ | $0.809_{\pm 0.003}$ | $\mathbf{0.766}_{\pm 0.008}$ |
| MCD | $0.525_{\pm 0.025}$ | $0.328_{\pm 0.048}$ | $0.472_{\pm 0.015}$ | $0.429_{\pm 0.007}$ | $0.024_{\pm 0.021}$ | $0.768_{\pm 0.009}$ | $0.738_{\pm 0.004}$ |
| | \multicolumn Ki-67 | | | | | | |
| | Pos. | Neg. | Stroma | | | Det. | Dice |
| Ours | $0.544_{\pm 0.151}$ | $0.655_{\pm 0.105}$ | $0.432_{\pm 0.053}$ | | | $0.809_{\pm 0.042}$ | $0.838_{\pm 0.031}$ |
| Ours w | $\mathbf{0.598}_{\pm 0.136}$ | $0.683_{\pm 0.069}$ | $0.461_{\pm 0.074}$ | | | $0.819_{\pm 0.042}$ | $0.827_{\pm 0.038}$ |
| Base | $0.531_{\pm 0.186}$ | $0.683_{\pm 0.076}$ | $0.437_{\pm 0.065}$ | | | $0.809_{\pm 0.035}$ | $0.825_{\pm 0.045}$ |
| DE | $0.574_{\pm 0.162}$ | $\mathbf{0.688}_{\pm 0.103}$ | $\mathbf{0.476}_{\pm 0.070}$ | | | $\mathbf{0.822}_{\pm 0.038}$ | $\mathbf{0.845}_{\pm 0.032}$ |
| MCD | $0.553_{\pm 0.067}$ | $0.626_{\pm 0.059}$ | $0.390_{\pm 0.075}$ | | | $0.797_{\pm 0.045}$ | $0.804_{\pm 0.043}$ |

this baseline, we apply two minor modifications: (i) Gaussian centroid maps are scaled by a factor of 100 to improve numerical stability (Xie et al., 2018); and (ii) training is performed for 200 epochs with constant learning rates ($2 \times 10^{-4}$ for PanNuke and $1 \times 10^{-4}$ for Ki-67) and batch sizes of 64 and 8, respectively. For centroid uncertainty, we use fixed weights $\lambda_{\mathrm{mass}} = 0.6$ and $\lambda_{\mathrm{peak}} = 0.3$ to form the combined score $u_{\mathrm{cent}}$. We include three segmentation-uncertainty baselines: (i) the original DualU-Net using Shannon entropy of the softmax output, (ii) Monte Carlo Dropout (MCD), implemented by applying spatial dropout ($p = 0.1$) after the last two blocks of the segmentation decoder and computing uncertainty over $T = 30$ stochastic forward passes, and (iii) a ten-model deep ensemble (DE) using the entropy of the ensemble-averaged predictions. We focus on MCD and DE as uncertainty baselines that can be integrated into the DualU-Net architecture with minimal structural changes. We consider two evidential variants: *Ours* with unweighted Dice and loss weights $\lambda_{\mathrm{seg}} = 1$, $\lambda_{\mathrm{dice}} = 0.4$, $\lambda_{\mathrm{cent}} = 0.7$, $\lambda_{\mathrm{kl}} = 0.4$, and *Ours w* with class-weighted Dice and $\lambda_{\mathrm{kl}} = 0.2$, both using a 40-epoch warm-up for $\lambda_{\mathrm{kl}}$. All hyperparameters have been selected on PanNuke validation folds and reused on Ki-67 without further tuning.

**Performance evaluation.** Performance results are reported in Table 1. Using paired two-sided $t$-tests across folds, we observe no statistically significant differences between our evidential approaches (*Ours*, *Ours w*) and the three considered baselines (Base, DE, and MCD) for any of the primary metrics, including Detection F1, Dice, and per-class F1 scores ($p > 0.05$ in all cases). A significant difference appears only for the rare *Necrotic* class in PanNuke, where *Ours w* achieves higher performance than *Ours* ($p = 0.015$). No such exception is observed on Ki-67, where no statistically significant differences are found for any metric or method pair.

**Evaluation metrics.** We evaluate uncertainty quality using Adaptive Calibration Error (ACE) (Nixon et al., 2019) and its maximum (MCE), as well as Adaptive UCE (A-UCE) and its maximum (M-UCE) using quantile-based binning (Laves et al., 2019). Error–uncertainty

Table 2: Quantitative uncertainty evaluation on PanNuke and Ki-67. *Left:* segmentation-head uncertainty and calibration results (EDL head) compared with Deep Ensembles (DE), Monte Carlo Dropout (MCD) and the deterministic DualU-Net baseline. *Right:* centroid-head uncertainty results. Complete centroid histograms and eCDF plots in Appendix E.

| M | UM | ACE ↓ | MCE ↓ | A-UCE ↓ | M-UCE ↓ | KS ↑ | AUROC ↑ |
|---|---|---|---|---|---|---|---|
| | | | | PanNuke | | | |
| *Ours* | $u_{\text{ale}}$ | $\mathbf{0.061}_{\pm 0.004}$ | $0.289_{\pm 0.010}$ | $0.157_{\pm 0.010}$ | $0.326_{\pm 0.025}$ | $0.392_{\pm 0.003}$ | $0.759_{\pm 0.003}$ |
| | $u_{\text{epi}}$ | $\mathbf{0.061}_{\pm 0.004}$ | $0.289_{\pm 0.010}$ | $0.100_{\pm 0.004}$ | $0.251_{\pm 0.017}$ | $0.392_{\pm 0.003}$ | $0.759_{\pm 0.003}$ |
| | $u_{\text{vac}}$ | $\mathbf{0.061}_{\pm 0.004}$ | $0.289_{\pm 0.010}$ | $\mathbf{0.054}_{\pm 0.004}$ | $0.246_{\pm 0.016}$ | $0.391_{\pm 0.003}$ | $0.758_{\pm 0.003}$ |
| *Ours w* | $u_{\text{ale}}$ | $0.095_{\pm 0.003}$ | $0.383_{\pm 0.005}$ | $0.175_{\pm 0.005}$ | $0.382_{\pm 0.008}$ | $0.442_{\pm 0.005}$ | $0.791_{\pm 0.002}$ |
| | $u_{\text{epi}}$ | $0.095_{\pm 0.003}$ | $0.383_{\pm 0.005}$ | $0.113_{\pm 0.002}$ | $0.333_{\pm 0.002}$ | $\mathbf{0.442}_{\pm 0.005}$ | $\mathbf{0.796}_{\pm 0.003}$ |
| | $u_{\text{vac}}$ | $0.095_{\pm 0.003}$ | $0.383_{\pm 0.005}$ | $0.080_{\pm 0.003}$ | $0.321_{\pm 0.003}$ | $0.441_{\pm 0.005}$ | $0.796_{\pm 0.003}$ |
| Base | $u_s$ | $0.234_{\pm 0.004}$ | $0.417_{\pm 0.027}$ | $0.198_{\pm 0.004}$ | $0.353_{\pm 0.027}$ | $0.287_{\pm 0.016}$ | $0.692_{\pm 0.010}$ |
| DE | | $0.131_{\pm 0.001}$ | $\mathbf{0.220}_{\pm 0.019}$ | $0.085_{\pm 0.001}$ | $\mathbf{0.159}_{\pm 0.013}$ | $0.344_{\pm 0.006}$ | $0.721_{\pm 0.003}$ |
| MCD | | $0.136_{\pm 0.014}$ | $0.194_{\pm 0.027}$ | $0.051_{\pm 0.017}$ | $0.088_{\pm 0.025}$ | $0.144_{\pm 0.071}$ | $0.602_{\pm 0.047}$ |
| | | | | Ki67 | | | |
| *Ours* | $u_{\text{ale}}$ | $\mathbf{0.106}_{\pm 0.048}$ | $\mathbf{0.161}_{\pm 0.040}$ | $0.217_{\pm 0.080}$ | $0.287_{\pm 0.069}$ | $0.452_{\pm 0.147}$ | $0.786_{\pm 0.088}$ |
| | $u_{\text{epi}}$ | $\mathbf{0.106}_{\pm 0.048}$ | $\mathbf{0.161}_{\pm 0.040}$ | $\mathbf{0.096}_{\pm 0.046}$ | $\mathbf{0.173}_{\pm 0.068}$ | $0.450_{\pm 0.146}$ | $0.787_{\pm 0.088}$ |
| | $u_{\text{vac}}$ | $\mathbf{0.106}_{\pm 0.048}$ | $\mathbf{0.161}_{\pm 0.040}$ | $0.111_{\pm 0.036}$ | $0.175_{\pm 0.027}$ | $0.446_{\pm 0.148}$ | $0.786_{\pm 0.089}$ |
| *Ours w* | $u_{\text{ale}}$ | $0.132_{\pm 0.053}$ | $0.220_{\pm 0.056}$ | $0.201_{\pm 0.086}$ | $0.258_{\pm 0.078}$ | $0.470_{\pm 0.156}$ | $0.796_{\pm 0.090}$ |
| | $u_{\text{epi}}$ | $0.132_{\pm 0.053}$ | $0.220_{\pm 0.056}$ | $0.122_{\pm 0.095}$ | $0.222_{\pm 0.123}$ | $\mathbf{0.471}_{\pm 0.157}$ | $\mathbf{0.796}_{\pm 0.090}$ |
| | $u_{\text{vac}}$ | $0.132_{\pm 0.053}$ | $0.220_{\pm 0.056}$ | $0.131_{\pm 0.059}$ | $0.195_{\pm 0.086}$ | $0.471_{\pm 0.159}$ | $0.796_{\pm 0.090}$ |
| Base | $u_s$ | $0.286_{\pm 0.153}$ | $0.430_{\pm 0.120}$ | $0.207_{\pm 0.144}$ | $0.226_{\pm 0.119}$ | $0.252_{\pm 0.113}$ | $0.663_{\pm 0.071}$ |
| DE | | $0.159_{\pm 0.120}$ | $0.283_{\pm 0.179}$ | $0.112_{\pm 0.065}$ | $0.252_{\pm 0.108}$ | $0.311_{\pm 0.126}$ | $0.690_{\pm 0.076}$ |
| MCD | | $0.203_{\pm 0.141}$ | $0.325_{\pm 0.179}$ | $0.121_{\pm 0.076}$ | $0.214_{\pm 0.137}$ | $0.226_{\pm 0.135}$ | $0.633_{\pm 0.135}$ |

| M | UM | KS ↑ | AUROC ↑ |
|---|---|---|---|
| | | PanNuke | |
| *Ours* | $u_{\text{cent}}$ | $0.429_{\pm 0.006}$ | $0.782_{\pm 0.003}$ |
| | $u_{\text{mass}}$ | $0.410_{\pm 0.005}$ | $0.767_{\pm 0.003}$ |
| | $u_{\text{peak}}$ | $0.338_{\pm 0.025}$ | $0.712_{\pm 0.016}$ |
| *Ours w* | $u_{\text{cent}}$ | $\mathbf{0.461}_{\pm 0.010}$ | $\mathbf{0.801}_{\pm 0.009}$ |
| | $u_{\text{mass}}$ | $0.448_{\pm 0.007}$ | $0.787_{\pm 0.009}$ |
| | $u_{\text{peak}}$ | $0.361_{\pm 0.015}$ | $0.723_{\pm 0.008}$ |
| | | Ki67 | |
| *Ours* | $u_{\text{cent}}$ | $0.591_{\pm 0.113}$ | $0.862_{\pm 0.058}$ |
| | $u_{\text{mass}}$ | $0.575_{\pm 0.121}$ | $0.851_{\pm 0.063}$ |
| | $u_{\text{peak}}$ | $0.520_{\pm 0.096}$ | $0.823_{\pm 0.052}$ |
| *Ours w* | $u_{\text{cent}}$ | $\mathbf{0.612}_{\pm 0.092}$ | $\mathbf{0.875}_{\pm 0.047}$ |
| | $u_{\text{mass}}$ | $0.596_{\pm 0.099}$ | $0.863_{\pm 0.056}$ |
| | $u_{\text{peak}}$ | $0.543_{\pm 0.058}$ | $0.843_{\pm 0.033}$ |

separability is quantified using the Kolmogorov–Smirnov (KS) statistic (Tan et al., 2025c) and AUROC, computed between continuous uncertainty values and binary correctness indicators. Calibration metrics (ACE, MCE, A-UCE, M-UCE) are reported only for the segmentation head, whose evidential formulation yields probabilistic class predictions. For the centroid head, uncertainty derives from geometric cues rather than calibrated probabilities; accordingly, only KS and AUROC are evaluated, as these measure how well uncertainty ranks correct versus incorrect detections.

**Segmentation uncertainty.** Table 2 (left) summarizes calibration and error-separation metrics for segmentation-head uncertainties. Across both datasets, the evidential formulation (*Ours* and *Ours w*) consistently improves the separation between correct and incorrect predictions. On PanNuke, all evidential uncertainties achieve substantially higher KS and AUROC than the deterministic baseline, with improvements that are highly statistically significant ($p < 10^{-6}$). Compared to MC Dropout, both evidential variants attain significantly higher KS and AUROC ($p < 0.05$), while differences with Deep Ensembles are not statistically significant ($p > 0.05$). The three evidential uncertainty measures behave similarly, with no statistically significant differences between them ($p > 0.1$). Distribution histograms and eCDF plots further confirm a clearer separation for evidential measures compared to all baselines (Figure 1). The weighted variant (*Ours w*) yields a statistically significant improvement over the unweighted model on PanNuke ($p < 0.05$). On Ki-67, both evidential variants outperform the deterministic baseline, Deep Ensembles, and MC Dropout in terms of KS and AUROC. Improvements over the baseline are statistically significant ($p < 10^{-4}$), while gains over Deep Ensembles are consistent but not statistically significant ($p > 0.1$). Compared to MC Dropout, *Ours w* achieves a statistically significant improvement in AUROC ($p < 0.05$), whereas KS differences do not reach significance

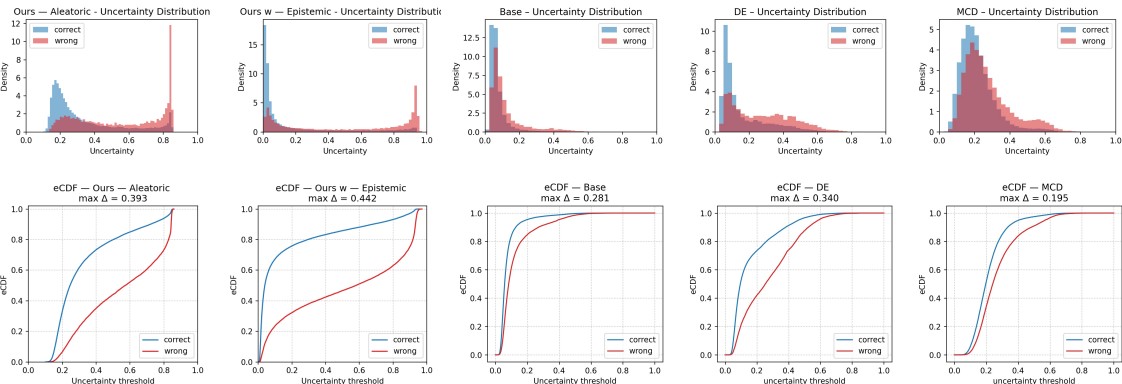

Figure 1: Segmentation-head uncertainty histograms (top) and eCDFs (bottom). Errors in red, correct instances in blue. Columns: *Ours*, *Ours w*, Base, DE, MCD. For evidential models we plot the best separator ($u_{\text{ale}}$ for *Ours*, $u_{\text{epi}}$ for *Ours w*). See additional histogram and eCDF analyses in Appendix D

($p > 0.1$); differences between *Ours* and MC Dropout are not statistically significant for either metric ($p > 0.1$). As on PanNuke, the three evidential uncertainties remain statistically indistinguishable ($p > 0.1$), and the weighted variant shows a consistent but not statistically significant improvement over the unweighted model.

**Segmentation calibration.** Across both datasets, our evidential variants show significantly improved calibration compared to the deterministic baseline, with all gains confirmed by statistical testing ($p < 10^{-4}$). Their calibration is statistically indistinguishable from Deep Ensembles and MC Dropout ($p > 0.1$), indicating ensemble-level performance. For MCE, both evidential variants significantly outperform the baseline, with stronger evidence for *Ours* ($p < 10^{-3}$) and a smaller but still significant effect for *Ours w* ($p < 0.05$), while Deep Ensembles remain the best-performing method. Compared to MC Dropout, the evidential variants exhibit higher miscalibration on PanNuke, with MC Dropout achieving significantly lower MCE and UCE-style errors ($p < 0.05$). On Ki-67, higher variance prevents statistically significant differences between methods ($p > 0.15$); nevertheless, the evidential models remain at least as well calibrated as Deep Ensembles, outperform the deterministic baseline, and achieve stronger calibration than MC Dropout in terms of ACE and MCE across folds ($p < 0.05$).

**Centroid uncertainty.** Table 2 (right) reports KS and AUROC for the centroid-head uncertainties ($u_{\text{peak}}, u_{\text{mass}}, u_{\text{cent}}$). On PanNuke, the proposed geometric cues provide clear discrimination, with the combined centroid score consistently outperforming the individual components. The weighted variant (*Ours w*) yields a statistically significant improvement in KS over the unweighted model ($p < 0.05$), while differences in AUROC remain within fold-to-fold variability ($p > 0.1$). Among the individual cues, the mass-based uncertainty is the most informative, followed by the peak-based cue, and their combination produces the strongest overall signal. On Ki-67, centroid uncertainties are even more discriminative. Both evidential variants achieve strong error separation across all centroid metrics, but no

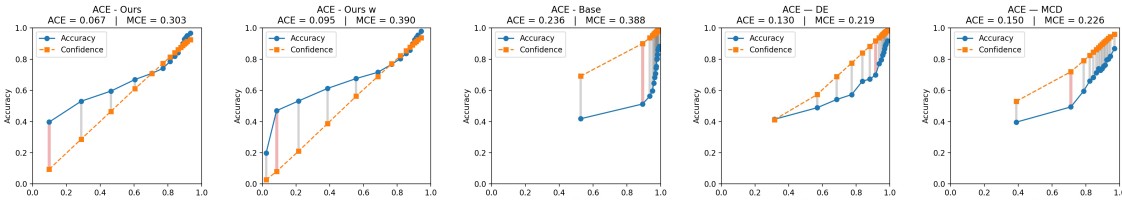

Figure 2: ACE plots for the segmentation head. Left to right: *Ours*, *Ours w*, Base, DE, MCD.

statistically significant differences are observed between *Ours* and *Ours w* ($p > 0.1$). As in PanNuke, the mass-based cue remains the most informative individual component, while the combined centroid uncertainty provides the most robust and stable separation.

**Qualitative results.** Figure 3 illustrates qualitative examples from a representative Pan-Nuke fold using the *Ours w* configuration. Across the examples, nuclei highlighted with high segmentation-head or centroid-head uncertainty consistently correspond to meaningful failure modes: clear classification mistakes, missed or imprecise detections, or instances that, despite being labeled as correct, exhibit ambiguous morphology or borderline staining and could warrant ground-truth revision.

## 5. Discussion and Conclusions

We have introduced an evidential formulation of DualU-Net that provides, in a single forward pass, two complementary uncertainty families: segmentation-driven evidential uncertainty (aleatoric, epistemic, vacuity) targeting classification errors, and centroid-derived geometric uncertainty (peak and mass) targeting detection and localisation errors. Together, they offer a unified decomposition of instance-level reliability that aligns with the two dominant failure modes in cell instance segmentation.

As shown in Table 1, incorporating EDL into the DualU-Net architecture does not degrade predictive performance: our evidential variants achieve comparable results to all considered baselines, with minor improvements over the deterministic Base model, but no statistically significant differences with respect to Base, DE, or MCD. Across PanNuke and Ki-67, the evidential scheme consistently outperforms the deterministic baseline and matches or surpasses DE and MCD in error separation, while being substantially more efficient. Although the three segmentation-head uncertainties differ qualitatively—aleatoric tending to produce higher intensities, epistemic and vacuity spanning wider dynamic ranges (Figure 1 and Appendix D)—their quantitative behaviour is statistically indistinguishable in terms of error discrimination (Table 2). In PanNuke, this alignment is visually evident (Figure 3): all three uncertainties assign high values to the same problematic nuclei, highlighting classification mistakes, false positives, or low-confidence predictions that merit inspection.

Regarding calibration, the evidential formulation improves mean calibration relative to DE and MCD, bringing predicted confidence closer to empirical correctness (Table 2). The

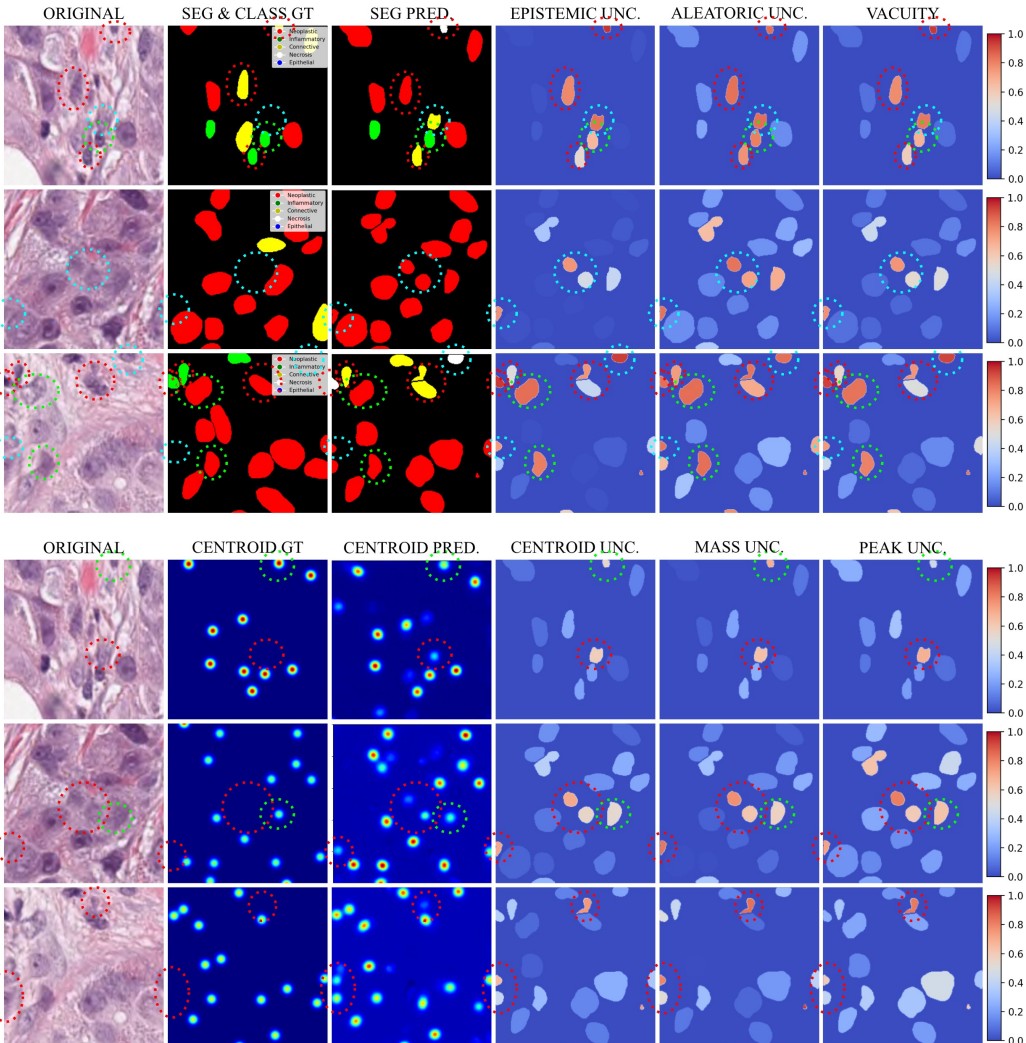

Figure 3: Qualitative uncertainty examples on PanNuke using the *Ours w* configuration. We show the original patch, ground-truth labels, predictions, and the three uncertainty measures for each head ($u_{\mathrm{epi}}$, $u_{\mathrm{ale}}$, $u_{\mathrm{vac}}$ for segmentation and $u_{\mathrm{cent}}$, $u_{\mathrm{mass}}$, $u_{\mathrm{peak}}$ for detection). For segmentation, red circles mark class-mismatch errors, blue false-positive nuclei, and green correctly predicted but ambiguous cases. For centroids, red circles highlight missed or imprecise detections, and green circles indicate correct detections with residual uncertainty. These examples illustrate how segmentation- and centroid-based uncertainties jointly identify unreliable instances.

strong calibration of DE and MCD is expected, as both average over multiple stochastic predictors, which inherently smooths confidence estimates. Despite relying on a single forward pass, our method achieves comparable or better calibration while maintaining similar uncertainty–error alignment at a much lower computational cost. Under-confidence at low predicted probabilities is an intrinsic effect of evidential modeling: when evidence is limited, vacuity dominates and the Dirichlet mean is drawn toward the uniform prior, reducing confidence even for correct predictions (Figure 2).

The class-weighted variant (*Ours w*) exposes a clear trade-off between rare-class performance and calibration. By amplifying gradients for underrepresented classes, class weighting promotes stronger evidence accumulation and improves fold-wise F1 scores, particularly for the Necrotic class in PanNuke (Table 1). At the same time, this reduces the regularising effect of the evidential KL term in low-sample regimes, allowing the model to become overly confident when evidence is scarce. Consequently, *Ours w* shows increased calibration error (e.g., higher MCE and M-UCE), reflecting a tension between enhancing rare-class sensitivity and maintaining conservative uncertainty estimates.

For the centroid head, the proposed Gaussian-based uncertainty measures are simple, interpretable, and computationally free at inference. Mass-based uncertainty is consistently the strongest cue, while peak uncertainty provides complementary information; their combination yields the best KS and AUROC values (Table 2). Qualitative examples confirm that high-uncertainty instances correspond to misdetections, poor localisations, or ambiguous annotations, demonstrating the practical interpretability of these geometric cues (Figure 3). The proposed centroid uncertainty cues rely on a Gaussian template with fixed standard deviation $\sigma$, which encodes an implicit prior on nucleus scale inherited from the original DualU-Net formulation. While the same $\sigma$ generalizes across PanNuke (H&E on 19 different tissue types) and Ki-67 (a different staining modality) datasets without retuning, applying the method to datasets with substantially different microns-per-pixel resolution or nucleus size distributions would require re-tuning this hyperparameter.

Importantly, all hyperparameters optimised on PanNuke transfer directly to Ki-67 without re-tuning, highlighting the cross-dataset generalisation of the evidential framework and its robustness under domain shift. The ability to surface uncertainty at inference time enables model introspection for pathologists and supports downstream applications such as active learning, quality control of annotations, and uncertainty-aware dataset curation.

Finally, we acknowledge recent work highlighting limitations of standard evidential formulations, including sensitivity to prior design choices and optimisation objectives that may induce over-confidence under certain conditions (Chen et al., 2024, 2025b). While our results demonstrate that a streamlined evidential formulation is already effective and competitive in a challenging multi-task instance segmentation setting, exploring such refinements within DualU-Net constitutes a natural avenue for future work.

To our knowledge, this is the first evidential instance segmentation model in a multi-task setting for digital pathology, demonstrating both methodological and practical value. Future work includes extending evidential modelling to centroid regression via Normal–Inverse–Gamma uncertainty (Amini et al., 2019), enabling a fully evidential DualU-Net architecture.

## Acknowledgments

This publication is part of the R&D&I project PID2023-148614OB-I00, funded by MICIU/AEI/10.13039/501100011033/ and by FEDER, EU. This research has also been funded by European Development Funds Regional, Programa operatiu FEDER de Catalunya 2014-2020 through the project DigiPatICS.

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

## Appendix A. Closed-Form Evidential Uncertainty for the Segmentation Head

**Dirichlet–Categorical evidential model.** For each pixel $x$, the segmentation head predicts Dirichlet concentration parameters $\boldsymbol{\alpha}(x) = (\alpha_1(x), \ldots, \alpha_K(x))$ with total evidence $S(x) = \sum_k \alpha_k(x)$. These induce a Dirichlet distribution $\mathbf{p}(x) \sim D(\mathbf{p} \mid \boldsymbol{\alpha}(x))$ over class probabilities, and a categorical prediction $\hat{y}(x) \sim \mathrm{Cat}(\mathbf{p}(x))$. Aleatoric and epistemic uncertainties are defined as

$$u_{\mathrm{ale}}(x) = \mathbb{E}_{\mathrm{Dir}}\big[\mathrm{Var}_{\mathrm{Cat}}\big(\hat{y} \mid \mathbf{p}\big)\big], \qquad u_{\mathrm{epi}}(x) = \mathrm{Var}_{\mathrm{Dir}}\big[\mathbb{E}_{\mathrm{Cat}}\big[\hat{y} \mid \mathbf{p}\big]\big]. \tag{8}$$

**Closed-form expressions.** For completeness, we derive the closed forms in Eq. (11) starting from the definitions in Section 3. Let $\boldsymbol{\alpha}(x) = (\alpha_1(x), \ldots, \alpha_K(x))$ and $S(x) = \sum_{k=1}^{K} \alpha_k(x)$, and denote $p_k$ the $k$-th component of $\mathbf{p}$.

- *Aleatoric uncertainty.* For a categorical variable with one-hot encoding, the conditional variance given $\mathbf{p}$ is

$$\mathrm{Var}_{\mathrm{Cat}}(\hat{y} \mid \mathbf{p}) = \sum_{k=1}^{K} p_k(1 - p_k) = \sum_{k=1}^{K} \big(p_k - p_k^2\big).$$

Taking the expectation under the Dirichlet prior,

$$u_{\mathrm{ale}}(x) = \mathbb{E}_{\mathrm{Dir}}\Big[\mathrm{Var}_{\mathrm{Cat}}(\hat{y} \mid \mathbf{p})\Big] = \sum_{k=1}^{K} \Big(\mathbb{E}[p_k] - \mathbb{E}[p_k^2]\Big).$$

Using standard Dirichlet moments,

$$\mathbb{E}[p_k] = \frac{\alpha_k(x)}{S(x)}, \qquad \mathbb{E}[p_k^2] = \frac{\alpha_k(x)\big(\alpha_k(x) + 1\big)}{S(x)\big(S(x) + 1\big)},$$

we obtain

$$\mathbb{E}[p_k] - \mathbb{E}[p_k^2] = \frac{\alpha_k(x)}{S(x)} - \frac{\alpha_k(x)\big(\alpha_k(x) + 1\big)}{S(x)\big(S(x) + 1\big)} = \frac{\alpha_k(x)\big(S(x) - \alpha_k(x)\big)}{S(x)\big(S(x) + 1\big)}.$$

Summing over $k$ gives

$$u_{\mathrm{ale}}(x) = \sum_{k=1}^{K} \frac{\alpha_k(x)\big(S(x) - \alpha_k(x)\big)}{S(x)\big(S(x) + 1\big)}. \tag{9}$$

- *Epistemic uncertainty.* The epistemic term is defined as the variability (under the Dirichlet prior) of the mean categorical prediction:

$$u_{\mathrm{epi}}(x) = \mathrm{Var}_{\mathrm{Dir}}\Big[\mathbb{E}_{\mathrm{Cat}}[\hat{y} \mid \mathbf{p}]\Big] = \sum_{k=1}^{K} \mathrm{Var}_{\mathrm{Dir}}(p_k),$$

where we sum the component-wise variances of $p_k$. For the Dirichlet,

$$\mathrm{Var}_{\mathrm{Dir}}(p_k) = \mathbb{E}[p_k^2] - \mathbb{E}[p_k]^2 = \frac{\alpha_k(x)\big(\alpha_k(x)+1\big)}{S(x)\big(S(x)+1\big)} - \left(\frac{\alpha_k(x)}{S(x)}\right)^2.$$

Bringing to a common denominator $S(x)^2\big(S(x)+1\big)$,

$$\mathrm{Var}_{\mathrm{Dir}}(p_k) = \frac{\alpha_k(x)\big(\alpha_k(x)+1\big)S(x) - \alpha_k^2(x)\big(S(x)+1\big)}{S(x)^2\big(S(x)+1\big)} = \frac{\alpha_k(x)\big(S(x)-\alpha_k(x)\big)}{S(x)^2\big(S(x)+1\big)}.$$

Summing over $k$ yields

$$u_{\mathrm{epi}}(x) = \sum_{k=1}^{K} \frac{\alpha_k(x)\big(S(x)-\alpha_k(x)\big)}{S(x)^2\big(S(x)+1\big)}. \tag{10}$$

Putting both together, we recover the compact expressions:

$$u_{\mathrm{ale}}(x) = \sum_{k=1}^{K} \frac{\alpha_k(x)\,(S(x)-\alpha_k(x))}{S(x)\,(S(x)+1)}, \qquad u_{\mathrm{epi}}(x) = \sum_{k=1}^{K} \frac{\alpha_k(x)\,(S(x)-\alpha_k(x))}{S(x)^2\,(S(x)+1)}. \tag{11}$$

## Appendix B. Theoretical Bounds and Normalization for Evidential Uncertainties

We derive here the analytic extrema used to normalize all uncertainties to $[0, 1]$. Let $\boldsymbol{\alpha}(x)$ be a $K$-class Dirichlet with total evidence $S(x) = \sum_k \alpha_k(x)$. To obtain meaningful and comparable bounds, we consider the extreme configurations of $\boldsymbol{\alpha}$ that maximise (or minimise) each uncertainty while remaining consistent with the evidential interpretation of the Dirichlet.

- *Bounds for aleatoric uncertainty.* Aleatoric uncertainty

$$u_{\text{ale}}(x) = \sum_{k=1}^{K} \frac{\alpha_k(S - \alpha_k)}{S(S+1)}$$

  quantifies intrinsic class ambiguity *given fixed evidence*. It is maximised when the classifier assigns equal expected class probabilities, i.e. when the Dirichlet is symmetric:

$$\alpha_1 = \cdots = \alpha_K = c, \qquad S = Kc.$$

  Substituting,

$$u_{\text{ale}} = \sum_{k=1}^{K} \frac{c(Kc - c)}{Kc(Kc+1)} = \frac{Kc(K-1)c}{Kc(Kc+1)} = \frac{K-1}{K} \cdot \frac{c}{c + \frac{1}{K}}.$$

  As $c \to \infty$ (high-evidence but fully ambiguous), this converges to

$$u_{\text{ale}}^{\max} = \frac{K-1}{K}.$$

  Regarding its minimum, it is 0, achieved when one class dominates ($\alpha_j \to S$, others $\to 0$).

- *Bounds for epistemic uncertainty.* Epistemic uncertainty

$$u_{\text{epi}}(x) = \sum_{k=1}^{K} \frac{\alpha_k(S - \alpha_k)}{S^2(S+1)}$$

  captures the variability of the Dirichlet mean under uncertain evidence. It is maximised when the model expresses *complete ignorance*, i.e. the Dirichlet concentration is at its weakest:

$$\alpha_1 = \cdots = \alpha_K = 1, \qquad S = K.$$

  Substituting,

$$u_{\text{epi}} = \sum_{k=1}^{K} \frac{1(K-1)}{K^2(K+1)} = K \frac{K-1}{K^2(K+1)} = \frac{K-1}{K(K+1)}.$$

  Any increase in evidence (larger $\alpha_k$) monotonically decreases $u_{\text{epi}}$, hence this is the theoretical maximum. Regarding its minimum, it is 0, achieved when one class dominates ($\alpha_j \to S$, others $\to 0$).

- *Bounds for vacuity.* Vacuity

$$u_{\text{vac}}(x) = \frac{K}{S(x)}$$

reflects the *absence of evidence.* Its minimum occurs when evidence is arbitrarily large $(S \to \infty)$:

$$u_{\text{vac}}^{\min} = 0.$$

Its maximum occurs when the evidence is minimal, i.e. $\alpha_k = 1$ for all classes, $S = K$:

$$u_{\text{vac}}^{\max} = \frac{K}{K} = 1.$$

Given any uncertainty value $u(x)$ with theoretical interval $[u_{\min}, u_{\max}]$, we map it to a common $[0, 1]$ scale via

$$\tilde{u}(x) = \frac{u(x) - u_{\min}}{u_{\max} - u_{\min}}.$$

This yields aligned and interpretable uncertainty scores across pixels, instances, uncertainty types, and datasets.

Table 3: Instance-level uncertainty evaluation under different aggregation strategies (mean, median, and sum) for Dirichlet parameters. Results are reported for the two evidential variants only (*Ours* and *Ours w*). Metrics and evaluation protocol follow the main paper.

| M | Agg. | UM | ACE ↓ | MCE ↓ | A-UCE ↓ | M-UCE ↓ | KS ↑ | AUROC ↑ |
|---|---|---|---|---|---|---|---|---|
| | | | | | PanNuke | | | |
| *Ours* | Mean | $u_{\text{ale}}$ | $\mathbf{0.061}_{\pm 0.004}$ | $\mathbf{0.289}_{\pm 0.010}$ | $0.157_{\pm 0.010}$ | $0.326_{\pm 0.025}$ | $0.392_{\pm 0.003}$ | $0.759_{\pm 0.003}$ |
| | Mean | $u_{\text{epi}}$ | $\mathbf{0.061}_{\pm 0.004}$ | $\mathbf{0.289}_{\pm 0.010}$ | $0.100_{\pm 0.004}$ | $0.251_{\pm 0.017}$ | $0.392_{\pm 0.003}$ | $0.759_{\pm 0.003}$ |
| | Mean | $u_{\text{vac}}$ | $\mathbf{0.061}_{\pm 0.004}$ | $\mathbf{0.289}_{\pm 0.010}$ | $\mathbf{0.054}_{\pm 0.004}$ | $\mathbf{0.246}_{\pm 0.016}$ | $0.391_{\pm 0.003}$ | $0.758_{\pm 0.003}$ |
| *Ours* | Sum | $u_{\text{ale}}$ | $0.061_{\pm 0.004}$ | $0.289_{\pm 0.010}$ | $0.185_{\pm 0.012}$ | $0.428_{\pm 0.019}$ | $0.392_{\pm 0.003}$ | $0.759_{\pm 0.003}$ |
| | Sum | $u_{\text{epi}}$ | $0.061_{\pm 0.004}$ | $0.289_{\pm 0.010}$ | $0.216_{\pm 0.004}$ | $0.468_{\pm 0.008}$ | $0.371_{\pm 0.005}$ | $0.743_{\pm 0.003}$ |
| | Sum | $u_{\text{vac}}$ | $0.061_{\pm 0.004}$ | $0.289_{\pm 0.010}$ | $0.216_{\pm 0.004}$ | $0.460_{\pm 0.007}$ | $0.350_{\pm 0.005}$ | $0.729_{\pm 0.003}$ |
| *Ours* | Median | $u_{\text{ale}}$ | $0.066_{\pm 0.003}$ | $0.312_{\pm 0.008}$ | $0.167_{\pm 0.004}$ | $0.360_{\pm 0.007}$ | $0.376_{\pm 0.002}$ | $0.752_{\pm 0.003}$ |
| | Median | $u_{\text{epi}}$ | $0.066_{\pm 0.003}$ | $0.312_{\pm 0.008}$ | $0.103_{\pm 0.004}$ | $0.283_{\pm 0.004}$ | $0.376_{\pm 0.002}$ | $0.753_{\pm 0.003}$ |
| | Median | $u_{\text{vac}}$ | $0.066_{\pm 0.003}$ | $0.312_{\pm 0.008}$ | $0.060_{\pm 0.002}$ | $0.277_{\pm 0.002}$ | $0.376_{\pm 0.002}$ | $0.753_{\pm 0.003}$ |
| *Ours w* | Mean | $u_{\text{ale}}$ | $0.095_{\pm 0.003}$ | $0.383_{\pm 0.005}$ | $0.175_{\pm 0.005}$ | $0.382_{\pm 0.008}$ | $0.442_{\pm 0.005}$ | $0.791_{\pm 0.002}$ |
| | Mean | $u_{\text{epi}}$ | $0.095_{\pm 0.003}$ | $0.383_{\pm 0.005}$ | $0.113_{\pm 0.002}$ | $0.333_{\pm 0.002}$ | $\mathbf{0.442}_{\pm 0.005}$ | $\mathbf{0.796}_{\pm 0.003}$ |
| | Mean | $u_{\text{vac}}$ | $0.095_{\pm 0.003}$ | $0.383_{\pm 0.005}$ | $0.080_{\pm 0.003}$ | $0.321_{\pm 0.003}$ | $0.441_{\pm 0.005}$ | $0.796_{\pm 0.003}$ |
| *Ours w* | Sum | $u_{\text{ale}}$ | $0.094_{\pm 0.003}$ | $0.383_{\pm 0.005}$ | $0.217_{\pm 0.005}$ | $0.499_{\pm 0.014}$ | $0.442_{\pm 0.005}$ | $0.795_{\pm 0.003}$ |
| | Sum | $u_{\text{epi}}$ | $0.094_{\pm 0.003}$ | $0.383_{\pm 0.005}$ | $0.245_{\pm 0.006}$ | $0.545_{\pm 0.008}$ | $0.429_{\pm 0.004}$ | $0.775_{\pm 0.001}$ |
| | Sum | $u_{\text{vac}}$ | $0.094_{\pm 0.003}$ | $0.383_{\pm 0.005}$ | $0.245_{\pm 0.006}$ | $0.528_{\pm 0.010}$ | $0.413_{\pm 0.002}$ | $0.764_{\pm 0.001}$ |
| *Ours w* | Median | $u_{\text{ale}}$ | $0.109_{\pm 0.004}$ | $0.418_{\pm 0.009}$ | $0.184_{\pm 0.004}$ | $0.416_{\pm 0.017}$ | $0.431_{\pm 0.004}$ | $0.783_{\pm 0.002}$ |
| | Median | $u_{\text{epi}}$ | $0.109_{\pm 0.004}$ | $0.418_{\pm 0.009}$ | $0.126_{\pm 0.002}$ | $0.370_{\pm 0.005}$ | $0.431_{\pm 0.004}$ | $0.791_{\pm 0.002}$ |
| | Median | $u_{\text{vac}}$ | $0.109_{\pm 0.004}$ | $0.418_{\pm 0.009}$ | $0.096_{\pm 0.005}$ | $0.359_{\pm 0.006}$ | $0.431_{\pm 0.004}$ | $0.791_{\pm 0.002}$ |

## Appendix C. Ablation Study on Instance-level Dirichlet Aggregation

This appendix analyzes the sensitivity of instance-level evidential uncertainty to the choice of pixel-wise pooling operation. We compare three aggregation strategies—mean, sum, and median—applied to pixel-level Dirichlet parameters within each watershed-derived cell instance. The ablation is performed for both *Ours* and *Ours w* variants using three cross-validation folds on PanNuke, while keeping the network, training protocol, and evaluation metrics unchanged. The goal is to assess whether the choice of pooling operation materially affects calibration, error–uncertainty separation, and interpretability of instance-level uncertainties.

Quantitative results are summarized in Table 3. Mean aggregation consistently yields the best performance across calibration (ECE, ACE, UCE), ranking-based metrics (AUROC), and distributional tests (KS). Sum aggregation leads to systematically degraded calibration, particularly for epistemic uncertainty and vacuity, while median aggregation performs closer to mean but with slightly weaker error–uncertainty separation. Paired $t$-tests across folds confirm that mean aggregation significantly outperforms sum aggregation for calibration-related metrics ($p < 0.01$ for ECE, UCE, and Adj-UCE) and significantly outperforms median aggregation for AUROC. No metric shows a statistically significant advantage for sum or median over mean aggregation.

The qualitative behavior underlying these trends is illustrated in Figures 4 and 5, which report instance-level histograms of epistemic uncertainty and vacuity. For sum aggregation (right panels), both quantities collapse toward zero for nearly all instances. This effect is caused by the growth of the total Dirichlet concentration $S$ with instance size, which suppresses epistemic uncertainty and vacuity regardless of prediction correctness. Although some error separation may remain, the resulting uncertainty values are poorly calibrated

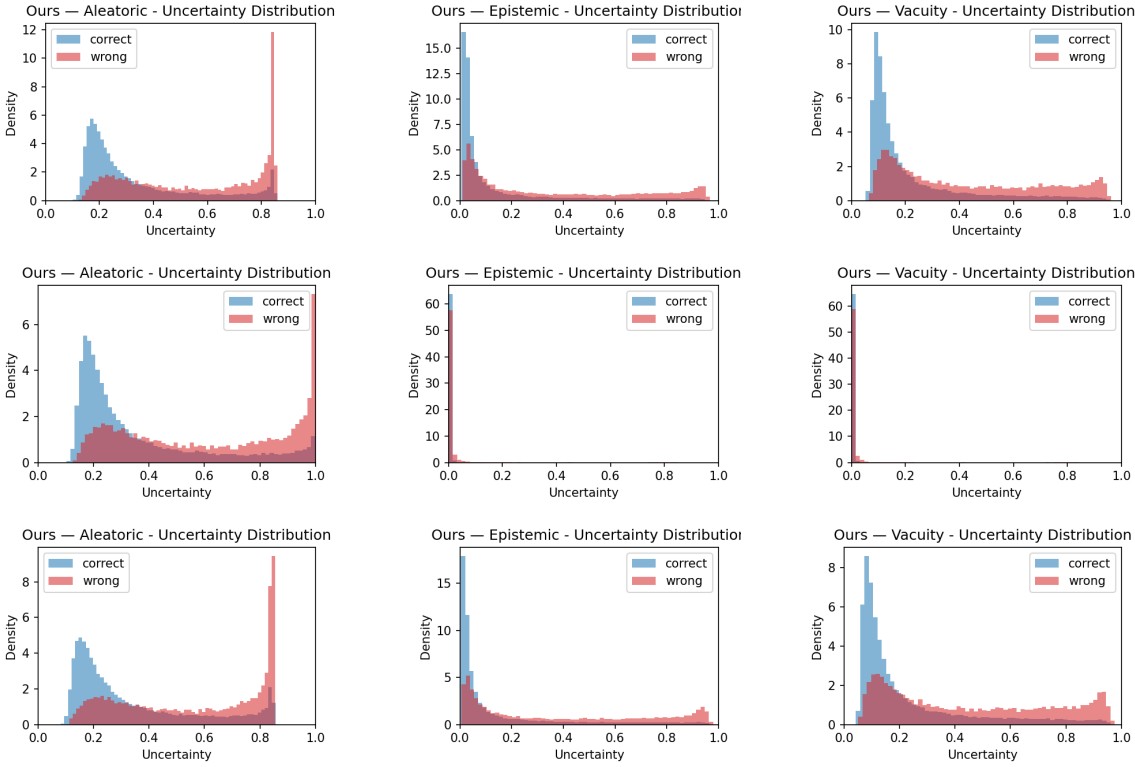

Figure 4: Instance-level histograms for segmentation uncertainties ($u_{\text{ale}}$, $u_{\text{epi}}$, $u_{\text{vac}}$) on Pan-Nuke, *Ours*. Rows, top to bottom, *mean, sum, median*.

and difficult to interpret. In contrast, median aggregation does not exhibit this collapse; however, its histograms are visually almost indistinguishable from those obtained with mean aggregation, for both *Ours* and *Ours w*, indicating no clear qualitative advantage over mean pooling.

Based on both quantitative and qualitative evidence, mean aggregation provides the most reliable instance-level uncertainty representation. It avoids the degenerate behavior induced by summation, preserves interpretability of epistemic uncertainty and vacuity, and achieves consistently better calibration and error–uncertainty separation than median pooling. These results justify the use of mean aggregation as a stable and size-invariant pooling operation for instance-level evidential uncertainty in the main paper.

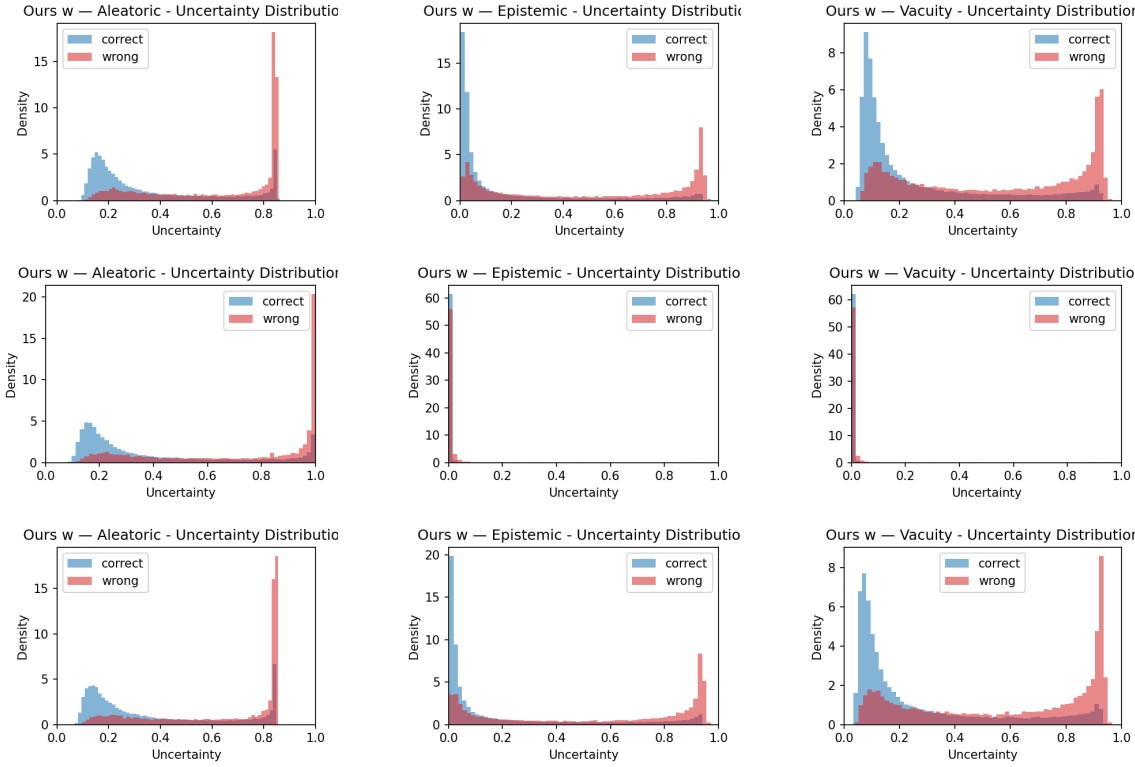

Figure 5: Instance-level histograms for segmentation uncertainties ($u_{\mathrm{ale}}$, $u_{\mathrm{epi}}$, $u_{\mathrm{vac}}$) on Pan-Nuke, *Ours w.* Rows, top to bottom, *mean, sum, median.*

## Appendix D. Additional segmentation-head uncertainty plots.

We provide full histogram and eCDF visualisations for all segmentation-head uncertainties $(u_{\mathrm{ale}}, u_{\mathrm{epi}}, u_{\mathrm{vac}})$ at the instance level, separately for PanNuke and Ki-67 and for both evidential variants. Specifically, Figures 6 and 7 show the results for *Ours* on PanNuke and Ki-67 respectively, while Figures 8 and 9 report the corresponding plots for *Ours w.* These visualisations complement the main paper results and consistently show strong separation between correct and incorrect nuclei across datasets and uncertainty types.

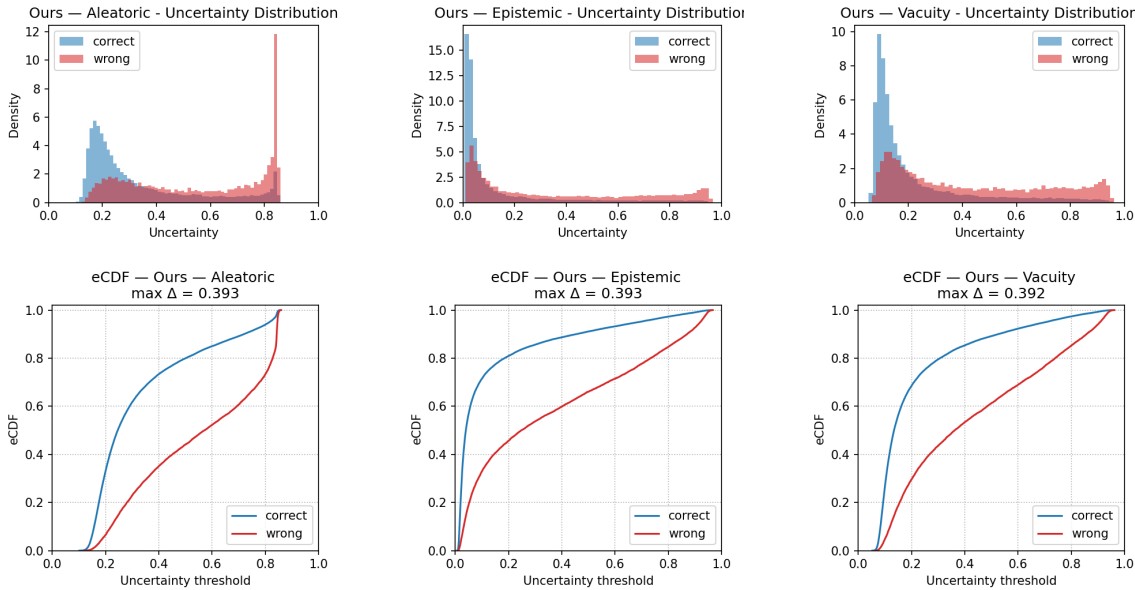

Figure 6: Instance-level histograms (top) and eCDFs (bottom) for segmentation uncertainties ($u_{\mathrm{ale}}$, $u_{\mathrm{epi}}$, $u_{\mathrm{vac}}$) on PanNuke, *Ours*. Errors in red, correct nuclei in blue.

## Appendix E. Additional Centroid Uncertainty Plots

We provide complete histogram and eCDF visualisations for all centroid-head uncertainties ($u_{\mathrm{peak}}$, $u_{\mathrm{mass}}$, and their linear combination $u_{\mathrm{cent}}$), separately for PanNuke and Ki-67 and for both evidential variants. Figures 10 and 11 correspond to *Ours*, and Figures 12 and 13 show the same plots for *Ours w*. Because centroid uncertainty arises from geometric cues rather than class probabilities, all evaluations are conducted at the *instance level*. Across datasets, errors consistently appear in the high-uncertainty tail, while correctly detected nuclei cluster at lower values.

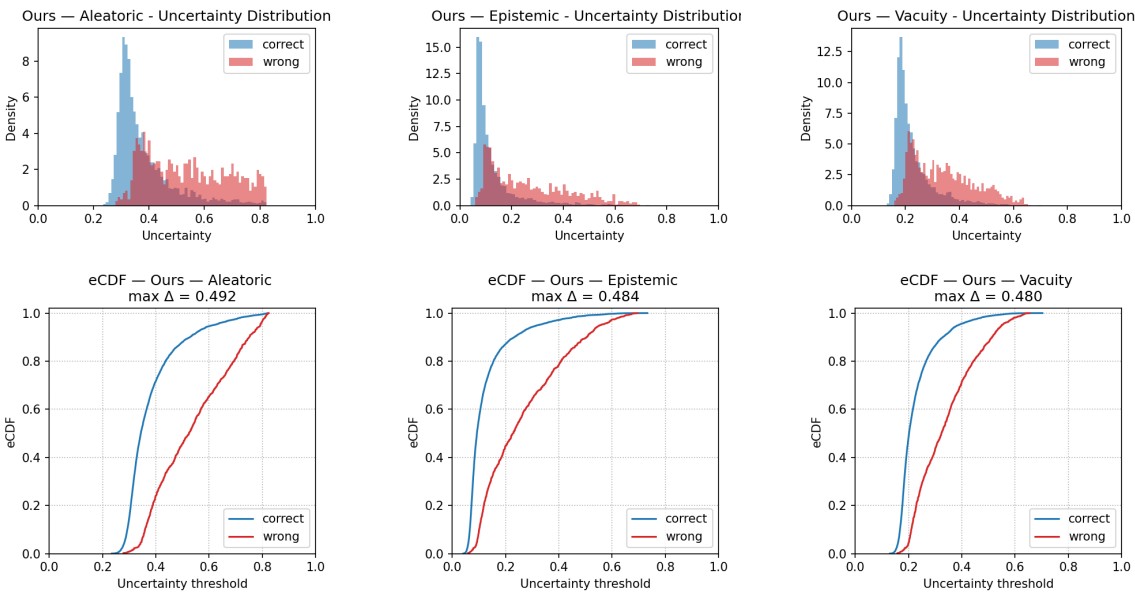

Figure 7: Instance-level segmentation uncertainty histograms and eCDFs for Ki-67, *Ours.* Errors in red, correct nuclei in blue.

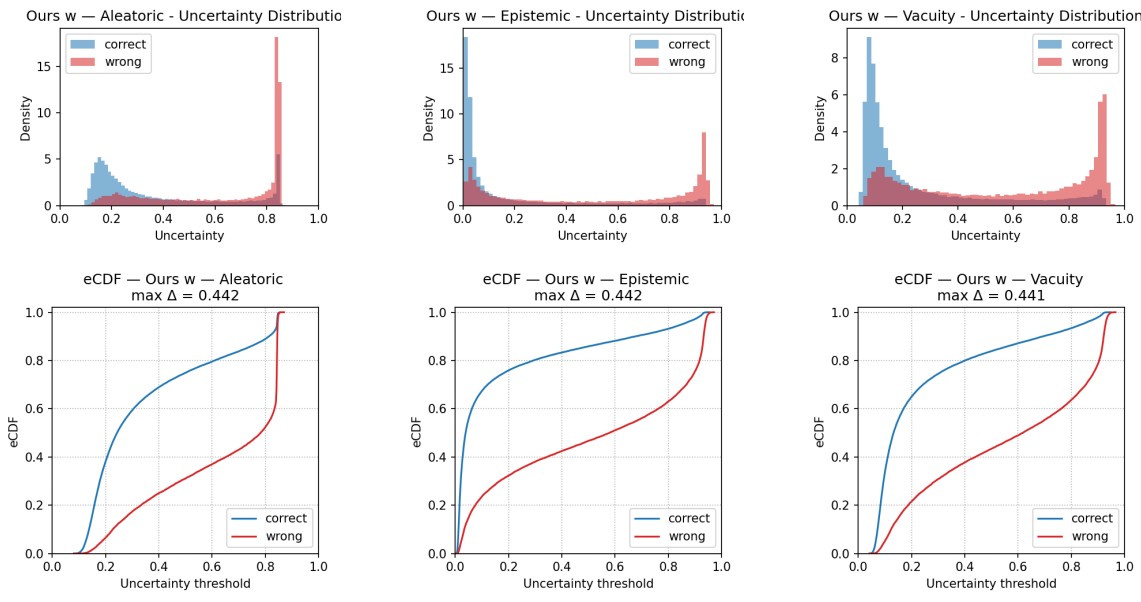

Figure 8: Instance-level segmentation uncertainty histograms and eCDFs for PanNuke, *Ours w.* Errors in red, correct nuclei in blue.

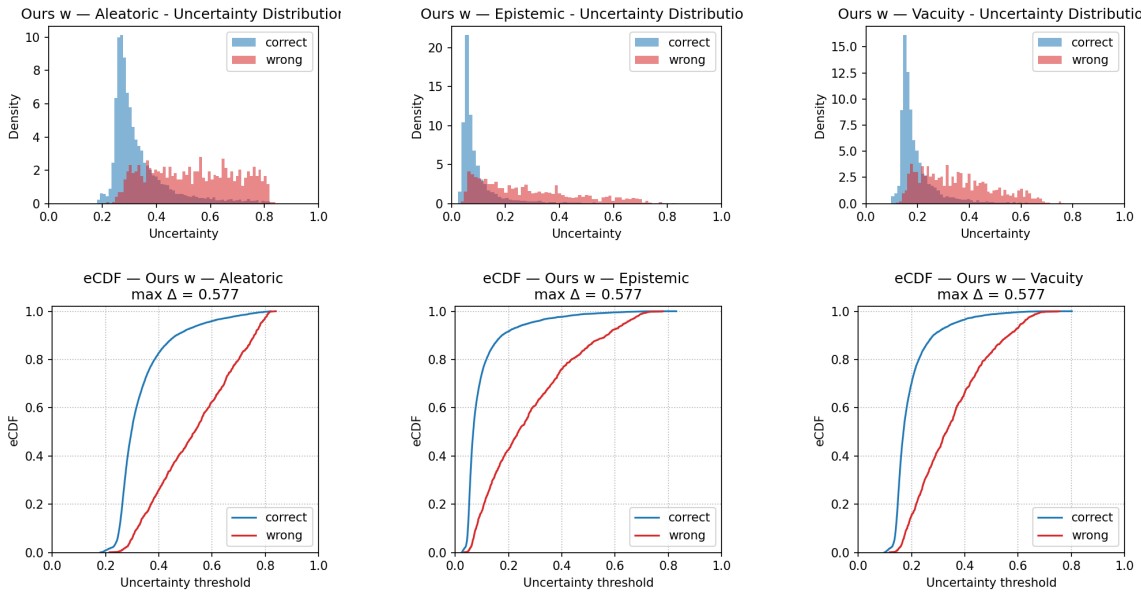

Figure 9: Instance-level segmentation uncertainty histograms and eCDFs for Ki-67, *Ours w*. Errors in red, correct nuclei in blue.

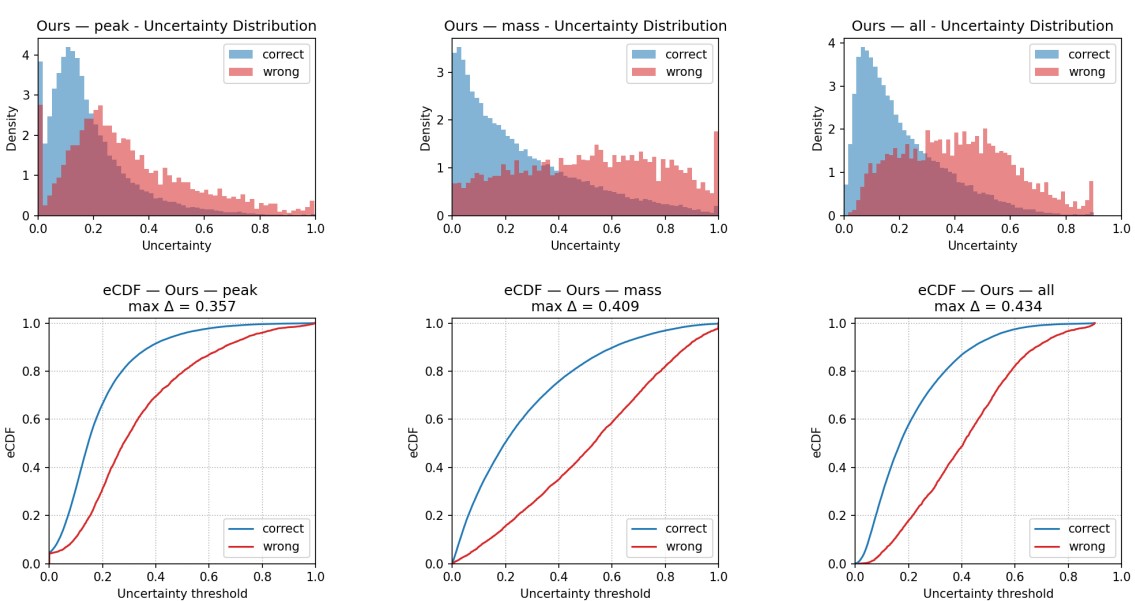

Figure 10: Instance-level centroid-head uncertainties on PanNuke for *Ours*. Top: histograms for $u_{\mathrm{peak}}, u_{\mathrm{mass}}, u_{\mathrm{cent}}$. Bottom: eCDFs. Errors in red, correct nuclei in blue.

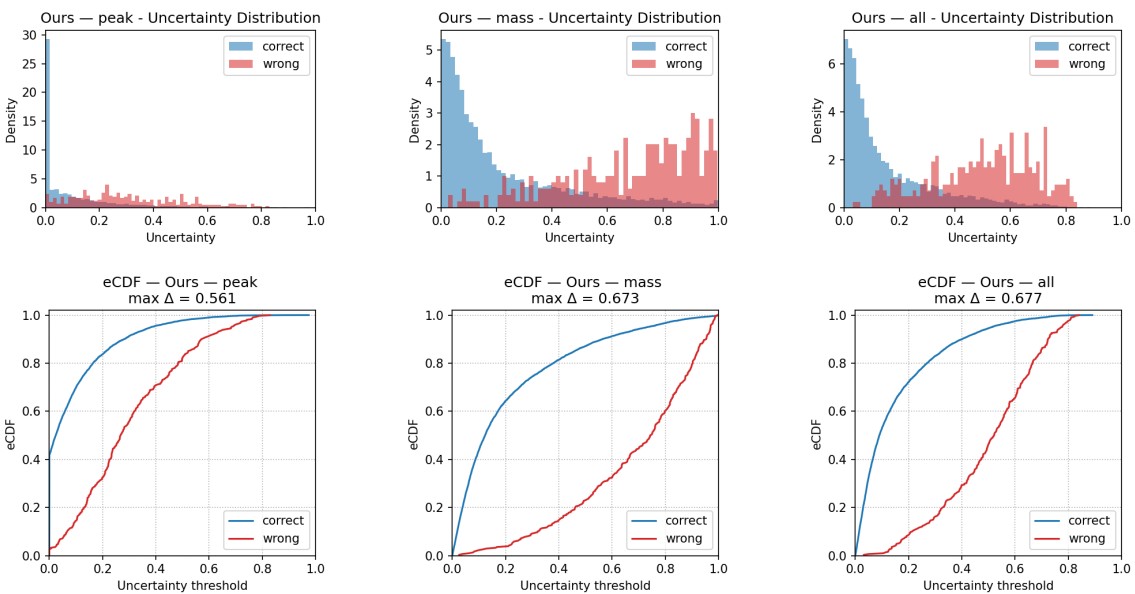

Figure 11: Instance-level centroid-head uncertainties on Ki-67 for *Ours*. Top: histograms. Bottom: eCDFs. Errors in red, correct nuclei in blue.

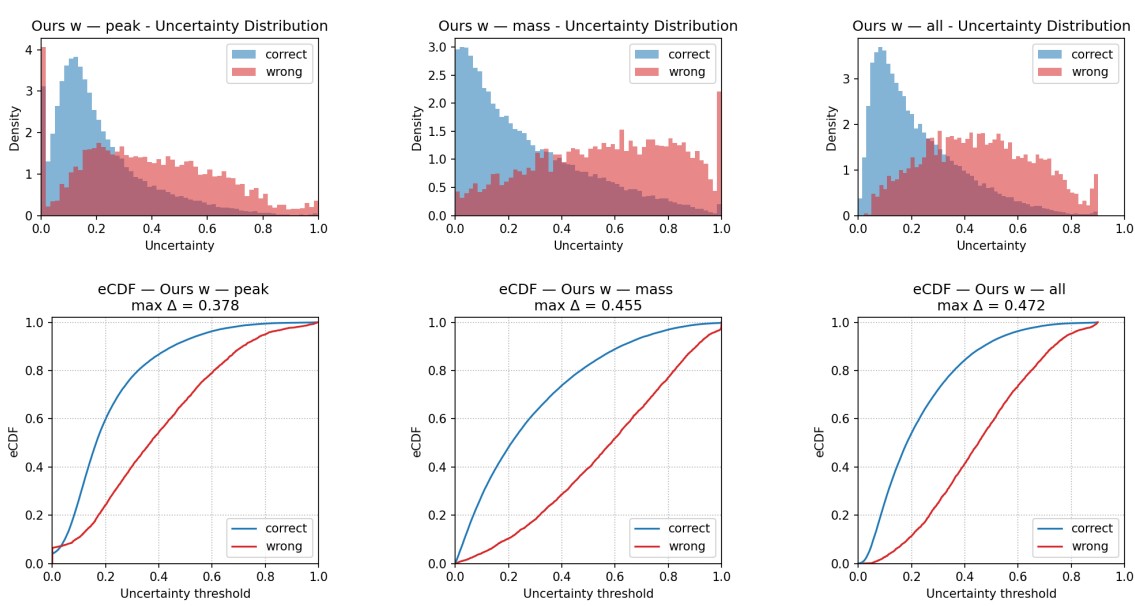

Figure 12: Instance-level centroid-head uncertainties on PanNuke for *Ours w*. Top: histograms. Bottom: eCDFs. Errors in red, correct nuclei in blue.

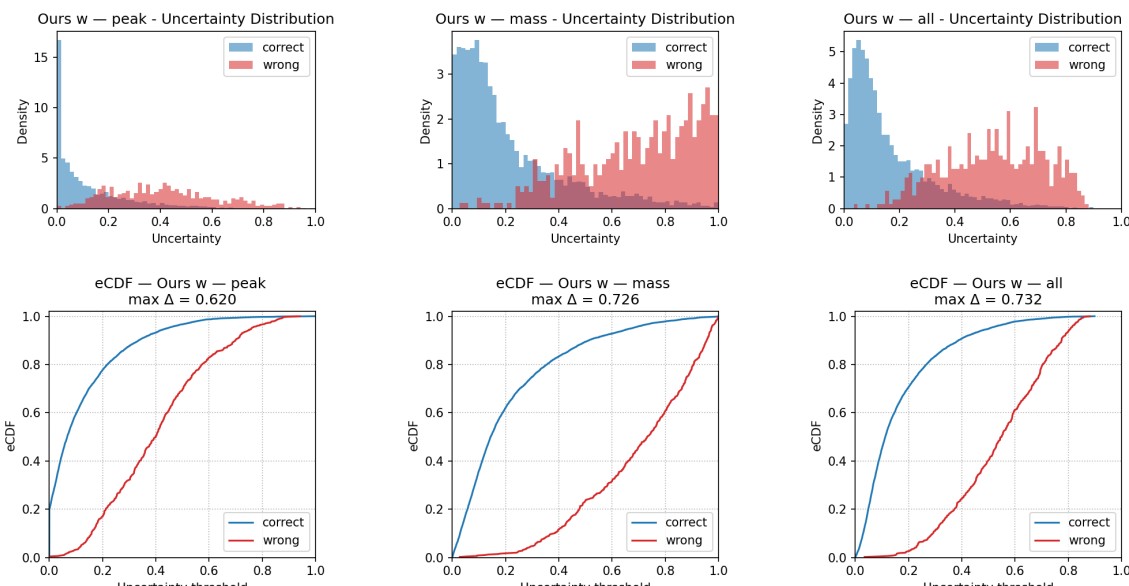

Figure 13: Instance-level centroid-head uncertainties on Ki-67 for *Ours w*. Top: histograms. Bottom: eCDFs. Errors in red, correct nuclei in blue.

