# OpenReview forum: "Evidential DualU-Net: Single-Pass Uncertainty for Cell Instance Segmentation"
_MIDL.io/2026/Conference — MIDL 2026 Poster_

### Official Review · Reviewer_iMCG · 2026-01-08

**Confidence:** 4
**Preliminary Rating:** 4
**Final Rating:** 4

**Summary:**

The paper proposes Evidential DualU-Net thereby extending DualU-Net to include uncertainty quantification for instance segmentation. Leveraging evidential deep learning, the framework provides classification uncertainty including epistemic, aleatoric and vacuity estimates as well as detection uncertainty. The single-pass characteristic of this framework makes it computationally more efficient than Deep Ensembles.

**Strengths:**

- Uncertainty evaluation is very thorough
- Very informative and well structured related work section
- All newly proposed methods are clearly motivated and derived from prior work
- The paper is well structured making it easy to follow
- Evaluation on two datasets showing that UQ generalises across datasets without re-tuning
- Clear evaluation of results including statistical tests

**Weaknesses:**

- Instance segmentation performance is not evaluated; while the inclusion of uncertainty is important, this should not come at a (great) cost to the instance segmentation performance
- Unclear what exactly is changed in the centroid regression head such that it predicts a Gaussian density map
- While this architecture builds on the DualU-Net and the original paper includes a graphic of the model architecture it would have been helpful to have an overview figure which clearly shows the changed architecture including the outputs of each head etc.
- Low quality / compressed plots

**Detailed Comments:**

- A short explanation or reference for watershed reconstruction would be helpful
- What is the rationale behind choosing DE and not prob. U-Net or PHiSeg?
- Which models are used for the DE (ResNet?)
- Baseline for centroid head: possibly object detection UQ methods?

**Justification Of Final Rating:**

I believe this paper provides valuable insights into incorporating uncertainty quantification into instance segmentation. The authors demonstrate that this can be achieved without a significant loss in segmentation performance.

**Justification Of The Preliminary Rating:**

Overall, the proposed idea is clinically relevant and supported by thorough experimental evidence, certain weaknesses remain. The authors should demonstrate that their addition of uncertainty quantification does not come at a (great) cost to the instance segmentation performance.

**Questions To Address In The Rebuttal:**

- Address the points raised in the detailed comments
- Instance segmentation performance
- Improve plot quality

---

> ### Author Response · Authors · 2026-01-25
>
> We thank the reviewer for the positive assessment and the constructive feedback. Below we address each concern in detail and clarify the corresponding changes made to the revised manuscript.
>
> 1. **Instance segmentation performance**: We agree that uncertainty estimation should not come at the cost of predictive performance. In the revised version, we have explicitly added a comprehensive performance evaluation in Section 4 (Performance evaluation), Table 2, reporting instance-level Detection F1, segmentation Dice, and per-class F1 scores for both PanNuke and Ki-67. Statistical significance is assessed using paired two-sided t-tests across folds. These results show that the proposed evidential variants achieve comparable performance to the deterministic baseline, Deep Ensembles, and MC Dropout, with no statistically significant degradation in any primary metric (p > 0.05 in all but one rare-class case). This directly demonstrates that uncertainty modeling does not adversely affect instance segmentation accuracy.
>
> 2. **Choice of uncertainty baselines**: We selected Deep Ensembles and MC Dropout as baselines because they can be directly applied to the DualU-Net architecture without architectural modification, allowing for a controlled comparison focused on uncertainty modeling rather than network design. In contrast, Probabilistic U-Net and PhiSeg require substantial architectural changes and different training objectives, which would confound the comparison and shift the scope of the study. Our goal is to assess how well-established uncertainty estimation strategies behave when applied to the same underlying instance segmentation model, rather than to benchmark across fundamentally different architectures. This rationale is now clarified in Section 4 (Experiments and implementation details).
>
> 3. **Centroid regression head and Gaussian density maps**: The centroid regression head is unchanged with respect to the original DualU-Net architecture, explained in Section 2 (Cell instance segmentation). This has now been clarified in Section 3 (Centroid-Head uncertainty). As in the original DualU-Net, the centroid decoder is trained to regress Gaussian density maps centered at nucleus locations using an MSE loss; no architectural modification is introduced for this head. The only architectural changes in our work concern the segmentation head, whose final layer is adapted to output non-negative evidence interpreted as Dirichlet parameters.
>
> 4. **Architecture overview figure**: We agree that an updated schematic could be helpful. A detailed overview of the DualU-Net architecture is available in the original DualU-Net paper. However, in Section 2 (Cell instance segmentation), we emphasize that the overall architecture is identical to DualU-Net, except for the final layer of the segmentation head, which now outputs evidential parameters instead of logits.
>
> 5. **Watershed reconstruction**: Following the reviewer’s suggestion, we have added a brief explanation for the watershed-based instance reconstruction in Section 2 (Cell instance segmentation), noting that we use the same post-processing pipeline as in the original DualU-Net, where watershed is applied over the predicted centroid and segmentation maps to obtain instance masks.
>
> 6. **Centroid-head uncertainty baselines**: For centroid localization, we focus on lightweight, geometry-based reliability cues derived directly from the predicted Gaussian maps. Existing object detection uncertainty methods typically rely on bounding boxes, anchor-based formulations, or probabilistic regression heads, which are not directly applicable to dense centroid regression and would again require architectural redesign. We therefore restrict comparisons to methods that preserve the original DualU-Net formulation and inference cost. As previously commented, this rationale is now clarified in Section 4 (Experiments and implementation details)
>
> 7. **Plot quality**: All figures have been regenerated at higher resolution, and compression artifacts have been removed in the revised manuscript.
>
> We hope these clarifications and additions address the reviewer’s concerns and improve the clarity and completeness of the manuscript.

---

> > ### Comment · Reviewer_iMCG · 2026-01-30
> >
> > Thank you for the clarifications regarding my aforementioned issues. I appreciate the segmentation performance evaluation.

---

### Official Review · Reviewer_ngeM · 2026-01-09

**Confidence:** 3
**Preliminary Rating:** 3
**Final Rating:** 4

**Summary:**

This paper extends DualU-Net to produce instance-level uncertainty without MC sampling or ensembles.   The key idea is to attach (i) a Dirichlet Deep Learning formulation to the segmentation head, enabling closed-form aleatoric / epistemic / vacuity maps, and (ii) two geometric consistency cues for the centroid head (peak and mass deviation under an assumed Gaussian template), producing an “uncertainty” score meant to rank localisation reliability.

**Strengths:**

Unlike Deep Ensembles (DE) or Monte Carlo (MC) Dropout, which require multiple forward passes to estimate uncertainty, Evidential DualU-Net extracts uncertainty from a single forward pass which is computational efficiency

By coupling Dirichlet-based evidential uncertainty for classification with novel geometric cues for detection, the framework uniquely addresses the dual nature of instance segmentation, explicitly separating semantic ambiguity from structural localization failures.

Evidential DualU-Net consistently outperforms deterministic baselines in calibration and error detection metrics, attaining statistical indistinguishability from computationally expensive Deep Ensembles.

By aggregating uncertainty at the clinically relevant instance level rather than the pixel level, the model provides interpretable reliability maps that flag specific error types and transfer to new datasets.

**Weaknesses:**

The segmentation head largely follows the canonical EDL approach (“evidence → Dirichlet α”, KL regularization, etc.) introduced by Sensoy et al[1], the authors also summarize in related work and note the EDL approach has been explored in prior semantic/biomedical segmentation. While the paper’s claimed novelty lies in extending evidential modeling to a multi-task, instance-level cell analysis pipeline.

- The instance level is by aggregation the pixel level map, which is not derived from a principled probabilistic composition rule, and the paper does not provide ablations or sensitivity analyses against alternative aggregations (e.g., sum/median/evidence-weighted or boundary-weighted schemes). Therefore, the methodological contribution beyond applying canonical EDL to the segmentation head appears limited.

Scale-Dependent Geometric Heuristics: The centroid uncertainty cues ($u_{peak}$ and $u_{mass}$) rely on a fixed Gaussian $\sigma$ to define the "ideal" mass $G_{max}$, effectively hard-coding a prior on nucleus scale.

- This dependency contradicts the claim of "generalization without retuning"4, as variations in scanner resolution or cell size would necessitate manual adjustment of $\sigma$ to prevent valid detections from being flagged as high-uncertainty errors. The lack of sensitivity analysis for $\sigma$ suggests these metrics function more as tuned consistency checks than robust uncertainty measures.

The paper interprets uncertainty by reading it directly from the Dirichlet parameters.

- However, recent analyses argue that many standard EDL objectives do not reliably recover Bayesian-style epistemic uncertainty[2], and other work proposes modified/relaxed EDL variants to mitigate overconfidence and improve behavior[3-4].

Ki-67 dataset is very small (4 patients, 52 tiles) which makes “generalizes without retuning” hard to conclude robustly.

[1]. Sensoy M, Kaplan L, Kandemir M. Evidential deep learning to quantify classification uncertainty[J]. Advances in neural information processing systems, 2018, 31.
[2] Shen, M., Ryu, J. J., Ghosh, S., Bu, Y., Sattigeri, P., Das, S., & Wornell, G. (2024). Are uncertainty quantification capabilities of evidential deep learning a mirage?. Advances in Neural Information Processing Systems, 37, 107830-107864.
[3] Chen, M., Gao, J., & Xu, C. (2024). R-edl: Relaxing nonessential settings of evidential deep learning. In The Twelfth International Conference on Learning Representations.
[4] Chen, M., Gao, J., & Xu, C. (2025). Revisiting essential and nonessential settings of evidential deep learning. IEEE Transactions on Pattern Analysis and Machine Intelligence.

**Detailed Comments:**

Instance-level aggregation of Dirichlet parameters.
- Consider adding a short justification and/or a small ablation showing robustness to pooling choice.

σ sensitivity is essential.
- A sensitivity curve: KS/AUROC of u_mass/u_peak/u_cent vs σ over a plausible range, and show whether the same σ works on PanNuke and Ki-67 might help the centroid head more soild.

**Justification Of Final Rating:**

The added ablation and discussion address my main concern about scale dependence and provide evidence that the proposed peak/mass cues are practically effective under the DualU-Net centroid-map formulation. In this application setting, treating σ as a fixed hyperparameter (inherited from DualU-Net) appears reasonable, and evaluating on 1024×1024 tiles (with patch-based processing) can still yield meaningful evidence on this dataset.

**Justification Of The Preliminary Rating:**

**Why not higher?**
The segmentation UQ is largely a standard EDL/Dirichlet add-on, and the epistemic/vacuity interpretation is still debated without targeted shift/data-scaling tests. More importantly, the centroid “uncertainty” is tightly tied to a Gaussian template scale (\sigma); without a sensitivity study, it may not transfer to new resolutions or nucleus sizes, weakening the “no retuning” generalisation claim.

**Why not lower?**
The approach is efficient (single-pass), conceptually clear (separating segmentation vs localisation failure modes), and the analysis is fairly complete for the paper’s scope, including strong deep-ensemble baselines plus separability and calibration evaluations showing near-ensemble performance on their datasets.

**Questions To Address In The Rebuttal:**

Please see comments and weakness.

---

> ### Author Response · Authors · 2026-01-25
>
> We thank the reviewer for the careful reading of our work, the positive assessment of its contributions, and for highlighting several important points that help clarify the scope and limitations of the proposed framework. Below we address each concern in detail.
>
> 1. **Instance-level aggregation of Dirichlet parameters**: We agree that instance-level aggregation of pixel-wise evidential predictions is not a principled probabilistic composition rule in the Bayesian sense, and that this point required clearer justification and empirical analysis. To address this, we have (i) expanded the methodological discussion in Section 3 (Segmentation-head evidential uncertainty) to explicitly clarify the interpretation of aggregation as a pooling operation over correlated pixel-level evidence rather than evidence fusion, and (ii) added a dedicated ablation study in Appendix C comparing different pooling strategies (mean, sum, and median). This study shows that summation leads to a collapse of epistemic uncertainty and vacuity due to the concentration parameter scaling with instance size, while median pooling yields behavior visually and quantitatively indistinguishable from mean pooling but with slightly inferior calibration. These results empirically support the choice of mean aggregation as a size-invariant and stable pooling operation for instance-level uncertainty estimation. References to this ablation are now explicitly included in the main text.
>
> 2. **Scale dependence and sensitivity of the centroid uncertainty cues (σ)**: We appreciate the reviewer’s emphasis on the role of the Gaussian scale parameter σ. Due to space limitations in the initial submission, we could not provide a detailed explanation of the DualU-Net design choices. We clarify that σ is not an uncertainty-specific heuristic, but a core hyperparameter of the DualU-Net architecture itself, as it defines how ground-truth centroid Gaussian maps are generated and therefore directly affects detection performance. This parameter was introduced and analyzed in the original DualU-Net paper following previous works for density estimation in cell and crowd counting [1], where its influence on detection accuracy was studied. In this work, we deliberately keep σ fixed to the value validated in DualU-Net, in order to isolate the contribution of evidential modeling and uncertainty estimation without altering the underlying detection setup. Importantly, σ affects the supervision of the centroid head rather than the uncertainty formulation per se: the proposed peak and mass-based uncertainty cues measure deviations from the predicted Gaussian structure learned during training. We have now explicitly discussed this distinction and its implications in Section 5 (Discussion and Conclusions). As noted there, if the model is applied at substantially different spatial resolutions or for markedly different nucleus sizes, σ would likely need to be re-estimated based on detection performance, rather than tuned for uncertainty behavior. In this study, we show that a single σ value generalizes across 19 tissues in PanNuke and across two different staining modalities (H&E and Ki-67).
>
> 3. **On limitations of evidential deep learning for epistemic uncertainty**: We thank the reviewer for pointing us to recent and highly relevant work questioning the Bayesian interpretation of uncertainty in standard EDL formulations. We found these references very insightful and have incorporated them into the revised manuscript. We now explicitly acknowledge in Section 5 (Discussion and Conclusions) that EDL-based epistemic and vacuity measures should be interpreted as model-internal uncertainty proxies rather than strict Bayesian epistemic uncertainty, and we position our contribution accordingly. Our empirical evaluation focuses on calibration and error separation, and this limitation is now clearly stated in Section 5 (Discussion and Conclusions).
>
> 4. **Ki-67 dataset size and generalization claims**: We agree that the Ki-67 dataset is limited in terms of patient count. However, each Ki-67 tile is substantially larger (1024×1024) than PanNuke patches (256×256), with a single tile corresponding to 16 PanNuke patches. As a result, the absolute number of evaluated instances is comparable, and the dataset provides a meaningful test of cross-stain generalization.
>
> Overall, we believe these clarifications and additions significantly strengthen the methodological rigor of the paper while keeping the focus on the central contribution: efficient, single-pass instance-level uncertainty estimation for cell segmentation. We thank the reviewer again for the constructive feedback, which has directly improved the quality and clarity of the manuscript.
>
> [1] Weidi Xie, J Alison Noble, and Andrew Zisserman. Microscopy cell counting and detection with fully convolutional regression networks. Computer methods in biomechanics and biomedical engineering: Imaging & Visualization, 6(3):283–292, 2018.

---

> > ### Comment · Reviewer_ngeM · 2026-01-31
> >
> > The added ablation and discussion address my main concern about scale dependence and provide evidence that the proposed peak/mass cues are practically effective under the DualU-Net centroid-map formulation. In this application setting, treating σ as a fixed hyperparameter (inherited from DualU-Net) appears reasonable, and evaluating on 1024×1024 tiles (with patch-based processing) can still yield meaningful evidence on this dataset.
> >
> > Overall, the revisions strengthen the paper’s support for an efficient, single-pass instance segmentation framework with useful instance-level reliability estimates. I will raise my overall score to 4.

---

### Official Review · Reviewer_3aLL · 2026-01-09

**Confidence:** 3
**Preliminary Rating:** 3
**Final Rating:** 3

**Summary:**

The paper proposes Evidential DualU-Net, a single-pass framework for cell instance segmentation along with uncertainty estimation. By integrating Evidential Deep Learning (EDL) into a multi-task architecture, the authors derive closed-form aleatoric, epistemic, and vacuity uncertainties for classification, alongside geometric cues for detection reliability. They evaluated the model on PanNuke and Ki-67 datasets, the model demonstrates performance comparable to deep ensembles and improved calibration over a deterministic baseline.

**Strengths:**

1) Introduces the first evidential deep learning framework for multi-task cell instance segmentation in digital pathology, extending EDL to instance-level tasks with novel geometric cues for localization errors.
2) The method provides single-pass, closed-form uncertainty estimates (aleatoric, epistemic, vacuity), avoiding costly ensembles or sampling methods.
3) Achieves error separation (KS, AUROC) matching or exceeding deep ensembles and markedly better calibration (ACE, MCE) than deterministic baseline, at much lower computational cost.

**Weaknesses:**

1) The aggregation of Dirichlet parameters $\alpha$ via simple averaging in Eq. 5 needs a rigorous Bayesian justification. Dirichlet parameters represent evidence, which is additive in the underlying Gamma space - averaging alters this semantics and may distort uncertainty quantities such as vacuity, particularly near instance boundaries. The impact of this approximation is not theoretically analyzed or empirically ablated.
2) While compared to Deep Ensembles, the paper misses comparisons to other baseline uncertainty methods, such as MC Dropout and the latest Variational U-Nets. Additionally, the geometric centroid cues ($u_{cent}$) could have been benchmarked against simpler alternatives, such as the Shannon entropy of the predicted Gaussian maps, to better isolate their contribution.
3) The "Ours w" variant improves error separation but significantly degrades calibration(MCE increasing by approximately 32%), suggesting a conflict between class-weighting and evidential regularization - an explicit discussion or analysis of this trade-off would strengthen the paper.
4) The paper reports uncertainty metrics for the Deep Ensemble baseline, but does not include raw performance metrics (Dice, F1). This prevents a full assessment of the accuracy-speed trade-off.
5) Since instance-level uncertainty is calculated within the watershed-defined mask $\Omega_i$, if the watershed incorrectly merges two cells, the resulting $\bar{\alpha}$ will be an average of two different cell types. The paper does not explicitly evaluate how the model handles the uncertainty of association.

**Detailed Comments:**

Please refer to the Weaknesses section.

**Justification Of Final Rating:**

While the proposed framework offers a constructive and computationally efficient extension to the DualU-Net architecture, I am recommending a Borderline rating to reflect a few remaining empirical and methodological considerations. A broader comparison with current state-of-the-art Bayesian or probabilistic segmentation models would be helpful to fully contextualize the framework’s technical standing within the existing landscape of uncertainty estimation. Currently, the proposed approach yields results that are comparable to, yet generally trail, the Deep Ensemble (DE) baseline across several key metrics. Specifically, DE maintains a superior Segmentation Dice of 0.766 (versus 0.761) and significantly more refined calibration, with an MCE of 0.220 compared to the 0.383 observed in the class-weighted variant.

Regarding the generalization analysis, while the authors clarify that the high pixel-count of the Ki-67 tiles provides a substantial number of evaluated instances, the reliance on a cohort of 4 patients remains a point of interest. In digital pathology, increasing the instance count is valuable, but it may not fully substitute for broader cohort diversity in capturing the inter-patient staining and morphological variability common in clinical settings. Furthermore, while the authors provided an insightful discussion regarding the 32% calibration degradation in the weighted variant, this remains an inherent trade-off rather than a methodological resolution, potentially leading to overconfidence when predicting rare but critical cell types. While the single-pass efficiency is a clear strength, the work's impact would be further solidified by demonstrating its robustness through benchmarking against a wider range of established probabilistic frameworks on more diverse datasets.

**Justification Of The Preliminary Rating:**

The paper presents a well-motivated evidential framework and demonstrates promising results on challenging instance segmentation tasks. The single-pass uncertainty formulation is appealing and computationally efficient. However, as raised in the points above, several aspects require further attention. Addressing these points would strengthen both the methodological rigor and the empirical support of the paper.

**Questions To Address In The Rebuttal:**

Please refer to the Weaknesses section. Also, would suggest improving the resolution of the figures in the paper to improve readability.

---

> ### Author Response · Authors · 2026-01-25
>
> We thank the reviewer for their careful reading and constructive feedback. We have addressed the raised concerns as follows.
>
> 1. **Instance-level aggregation of Dirichlet parameters**: We agree that instance-level aggregation of pixel-wise evidential predictions is not a principled probabilistic composition rule in the Bayesian sense, and that this point required clearer justification and empirical analysis. To address this, we have (i) expanded the methodological discussion in Section 3 (Segmentation-head evidential uncertainty) to explicitly clarify the interpretation of aggregation as a pooling operation over correlated pixel-level evidence rather than evidence fusion, and (ii) added a dedicated ablation study in Appendix C comparing different pooling strategies (mean, sum, and median). This study shows that summation leads to a collapse of epistemic uncertainty and vacuity due to the concentration parameter scaling with instance size, while median pooling yields behavior visually and quantitatively indistinguishable from mean pooling but with slightly inferior calibration. These results empirically support the choice of mean aggregation as a size-invariant and stable pooling operation for instance-level uncertainty estimation. References to this ablation are now explicitly included in the main text.
> 3. **Baseline uncertainty methods**: We selected Deep Ensembles and we added MC Dropout as baselines because both can be applied directly to the same DualU-Net architecture without architectural modification, enabling a controlled comparison focused on uncertainty estimation rather than network design. In contrast, Variational U-Net’s like Probabilistic U-Net and PhiSeg require substantial architectural changes and different training objectives, which would confound the comparison and shift the scope of the study toward architectural differences rather than uncertainty modeling. Our goal is to assess how established uncertainty estimation strategies behave when applied to the same underlying instance segmentation model. We have now explicitly clarified this rationale and added MC Dropout as a baseline in Section 4 (Experiments and implementation details). Regarding the centroid head, a clarification is needed. The predicted Gaussian maps are the output of a regression task, not a probabilistic classification over discrete outcomes. As such, they do not define a normalized probability distribution, and Shannon entropy is not well-defined in this setting.
>
> 3. **Trade-off between “Ours” and “Ours w”.**: We explicitly discuss the observed trade-off between error separation and calibration for the class-weighted variant “Ours w” in Section 5 (Discussion and Conclusions). While class weighting improves sensitivity to rare classes and enhances error separability, it interacts with evidential regularization and leads to degraded calibration, as reflected by higher MCE and UCE values. This trade-off is now clearly acknowledged and discussed in the revised manuscript.
>
> 4. **Instance segmentation performance.**: We agree that uncertainty estimation should not come at the cost of predictive performance. Performance metrics were already considered in the initial submission but were only briefly reported due to space limitations. In the revised version, we have explicitly added a comprehensive performance evaluation in Section 4 (Performance evaluation), Table 2, reporting instance-level Detection F1, segmentation Dice, and per-class F1 scores for both PanNuke and Ki-67. Statistical significance is assessed using paired two-sided t-tests across folds. These results show that the proposed evidential variants achieve comparable performance to the deterministic baseline, Deep Ensembles, and MC Dropout, with no statistically significant degradation in any primary metric (p > 0.05 in all but one rare-class case). This directly demonstrates that uncertainty modeling does not adversely affect instance segmentation accuracy.
>
> 5. **Watershed-induced instance merging.**
> Regarding the possible case where watershed reconstruction incorrectly merges two nuclei of different cell types, we note that this scenario would not silently mask errors. When pooling pixel-level Dirichlet parameters with mean aggregation, evidence from multiple classes is combined within the merged instance, resulting in increased uncertainty. Consequently, such errors are explicitly highlighted by the instance-level uncertainty estimates rather than suppressed. In our experiments, we did not observe cases where a single watershed-derived instance contained two clearly different cell types.
>
> Overall, we believe these additions substantially strengthen the methodological rigor of the paper and directly address the reviewer’s concerns regarding aggregation, baselines, and uncertainty interpretation.

---

### Author Rebuttal · Authors · 2026-01-25

**Rebuttal:**

We attach the revised version of our manuscript, which has been carefully updated in response to the reviewers’ constructive comments. We sincerely thank all reviewers for their thorough evaluations, positive feedback, and insightful suggestions, which have helped us improve both the clarity and methodological rigor of the paper. In this revision, all newly added or substantially modified text is highlighted in green to facilitate identification of the changes and to guide the reader through the updates. We believe that these revisions adequately address the main concerns raised during the review process and further strengthen the contribution of our work.

**Supporting Material:**

/attachment/108ec80db7c79cb0a931119adc29edd72eea4852.pdf

---

### Meta-Review · Area_Chair_tKTq · 2026-02-09

**Recommendation:** Accept (Poster)
**Confidence:** 4

**Metareview:**

The paper proposes Evidential DualU-Net by extending DualU-Net to include uncertainty quantification for cell instance segmentation. The authors integrate Evidential Deep Learning (EDL) into a multi-task architecture,and derive closed-form aleatoric, epistemic, and vacuity uncertainties for classification. The single-pass characteristic of this framework makes it computationally efficient.
 Most of the concerns raised by the reviewers have been addressed in the rebutall.

---

### Decision · Program_Chairs · 2026-02-14

Accept (Poster)